# Metabolic and epigenetic dysfunctions underlie the arrest of in vitro fertilized human embryos in a senescent-like state

Yang Yang[1]☯, Liyang Shi[2]☯, Xiuling Fu[2], Gang Ma[2], Zhongzhou Yang[3], Yuhao Li[2], Yibin Zhou[2], Lihua Yuan[1], Ye Xia[1], Xiufang Zhong[1], Ping Yin[1], Li Sun[2], Wuwen Zhang[1], Isaac A. Babarinde[2], Yongjun Wang[4,5], Xiaoyang Zhao[6,7,8], Andrew P. Hutchins[2]‡*, Guoqing Tong[1]‡*

1 Center for Reproductive Medicine, Shuguang Hospital Affiliated to Shanghai University of Traditional Chinese Medicine, Shanghai, China, 2 Shenzhen Key Laboratory of Gene Regulation and Systems Biology, Department of Biology, School of Life Sciences, Southern University of Science and Technology, Shenzhen, China, 3 BGI Genomics, BGI-Shenzhen, Shenzhen, China, 4 Longhua Hospital, Shanghai University of Traditional Chinese Medicine, Shanghai, China, 5 Key Laboratory of Theory and Therapy of Muscles and Bones, Ministry of Education, Shanghai, China, 6 State Key Laboratory of Organ Failure Research, Department of Developmental Biology, School of Basic Medical Sciences, Southern Medical University, Guangzhou, Guangdong, China, 7 Guangdong Key Laboratory of Construction and Detection in Tissue Engineering, Southern Medical University, Guangzhou, Guangdong, China, 8 Guangzhou Regenerative Medicine and Health Guangdong Laboratory (GRMH-GDL), Guangzhou, Guangdong, China

☯ These authors contributed equally to this work.
‡ These authors are joint senior authors on this work.
* andrewh@sustech.edu.cn (APH); drivftongguoqing@shutcm.edu.cn (GT)

**Data Availability Statement:** The authors confirm that all data underlying the findings are fully available without restriction. The datasets

## Abstract

Around 60% of in vitro fertilized (IVF) human embryos irreversibly arrest before compaction between the 3- to 8-cell stage, posing a significant clinical problem. The mechanisms behind this arrest are unclear. Here, we show that the arrested embryos enter a senescent-like state, marked by cell cycle arrest, the down-regulation of ribosomes and histones and down-regulation of MYC and p53 activity. The arrested embryos can be divided into 3 types. Type I embryos fail to complete the maternal-zygotic transition, and Type II/III embryos have low levels of glycolysis and either high (Type II) or low (Type III) levels of oxidative phosphorylation. Treatment with the SIRT agonist resveratrol or nicotinamide riboside (NR) can partially rescue the arrested phenotype, which is accompanied by changes in metabolic activity. Overall, our data suggests metabolic and epigenetic dysfunctions underlie the arrest of human embryos.

## Introduction

In vitro fertilization (IVF) has revolutionized the treatment of human fertility problems. However, a large number of human embryos fail to develop in vitro, and typically, only 30% of human embryos will progress to the blastocyst stage [1,2]. Human preimplantation embryos can arrest at all stages between the zygote and the blastocyst, and a large fraction irreversibly

supporting the conclusions of this article are available in the GSA (Genome Sequence Archive): HRA001406 under controlled access for human samples. The normalized gene expression matrix for all samples and genes/TEs used in the study and the raw tag count matrix of all samples (excluding resveratrol) used in this study for CytoTRACE analysis are available at https://figshare.com/articles/dataset/Human_embryo_normalized_gene_expression_data/19775992.

**Funding:** This work was supported by the National Key R&D Program of China (2018YFC1704300 to Y.J.W.), the National Natural Science Foundation of China (81070494 and 81170571 to G.Q.T, 81571442 to W.Z., and 31970589 to A.P.H.), the Shenzhen Innovation Committee of Science and Technology (JCYJ20200109141018712 to A.P.H. and ZDSYS20200811144002008 to the Shenzhen Key Laboratory of Gene Regulation and Systems Biology and to A.P.H.), and the Stable Support Plan Program of the Shenzhen Natural Science Fund (20200925153035002 to A.P.H.). The funders had no role in study design, data collection and analysis, decision to publish, or preparation of the manuscript.

**Competing interests:** The authors have declared that no competing interests exist.

**Abbreviations:** DE, differentially expressed; ERV, endogenous retrovirus; ESC, embryonic stem cell; GSEA, gene set enrichment analysis; HSC, hematopoietic stem cell; IVF, in vitro fertilized; MZT, maternal-to-zygotic transition; NR, nicotinamide riboside; PCA, principal component analysis; TE, transposable element; TF, transcription factor; TSS, transcription start site; ZGA, zygotic genome activation.

arrest between the 2-cell and 8-cell stages and remain un-compacted [2]. Several cellular mechanisms have been proposed to explain this arrest, specifically: failed zygotic genome activation (ZGA) [3], delayed maternal RNA clearance [4], reactive oxygen species causing endoplasmic reticulum stress [5,6], and aneuploidy [7]. Computational machine learning techniques can detect morphological patterns in microscope images of otherwise normal-appearing embryos that will later go on to arrest [1], suggesting the arrest mechanisms are active before they manifest. However, the cellular mechanism remains unclear.

In vitro developmental models of embryogenesis in other organisms has not brought clarity to this problem, as preimplantation development is divergent between species [8]. For example, ZGA mainly occurs at the 2-cell stage in mice, but in humans, there are 2 waves, a minor ZGA at the 2-cell stage and the major ZGA at the 8-cell stage [9]. Relatedly, in contrast to humans, some species have good in vitro developmental potential. For example, approximately 90% of mouse, approximately 80% of (monospermic) pig, approximately 70% of cat, and approximately 60% of *Macaca mulatta* embryos successfully develop to the blastocyst stage [10–12]. Conversely, humans are not the only species with poor in vitro embryonic developmental potential, only 25% to 30% of cattle and horse embryos will develop to a blastocyst [6,13,14]. However, it is unclear if the same mechanisms are active in other species. Human embryonic stem cells (ESCs) can be manipulated to form artificial blastocyst-like "blastoids" that mimic natural blastocysts [15,16]. Interestingly, blastoids are generated at low efficiency, which may reflect developmental problems inherent to natural blastocysts. However, blastoids cannot address pre-morula developmental arrest, as they model a later developmental stage, and it is unclear if the problems seen in blastoids are the same as preimplantation embryos. Ultimately, to investigate the arrest of human embryos, it is necessary to assay the problems directly.

In this study, we explored the transcriptomic basis behind the arrest of human embryos. A subset of the arrested embryos enter into a senescent-like state characterized by the up-regulation of p53, MYC, FOXO1, and the widespread down-regulation of ribosomes, histones, and translation initiation factors. We show that this senescent phenotype can be partially overcome using the antioxidant resveratrol and nicotinamide riboside (NR), and our data suggest that these 2 molecules activate the sirtuin family of acetyltransferases (SIRTs) to modulate metabolism. Modulation of SIRT activity leads to a reactivation of the arrested embryos and progression to a morula and early blastocyst.

## Results

### Gene expression of arrested human embryos after in vitro fertilization

Under typical IVF procedures, approximately 60% of embryos arrest (**Fig 1A**). We were interested in the class of embryos that arrest during development, but maintained a normal morphology and cell integrity, and did not show signs of disintegration. Embryos often arrest at either day 3 or day 4, postfertilization (**Fig 1A** and **S1 Data**). Day 4-arrested embryos reach the 8-cell stage but would fail to form a morula. The day 3–arrested embryos would undergo cleavage, but failed to reach the 8-cell stage (**Fig 1B** and **1C** and **S1 Data**). In this study, we focused on the day 3–arrested embryos. The embryos were left a further day to confirm no further development and no fragmentation or disintegration of the embryonic cells. Under these criteria, the embryos would be discarded in IVF procedures. It should be noted that, by this definition, approximately 3% of the day 3–arrested embryos can spontaneously recommence development and form a blastocyst (see later in the manuscript). However, prolonged in vitro culture of human embryos is deleterious for further development [2]; hence, we chose the minimal window between confirming the arrest of the embryos and preserving developmental

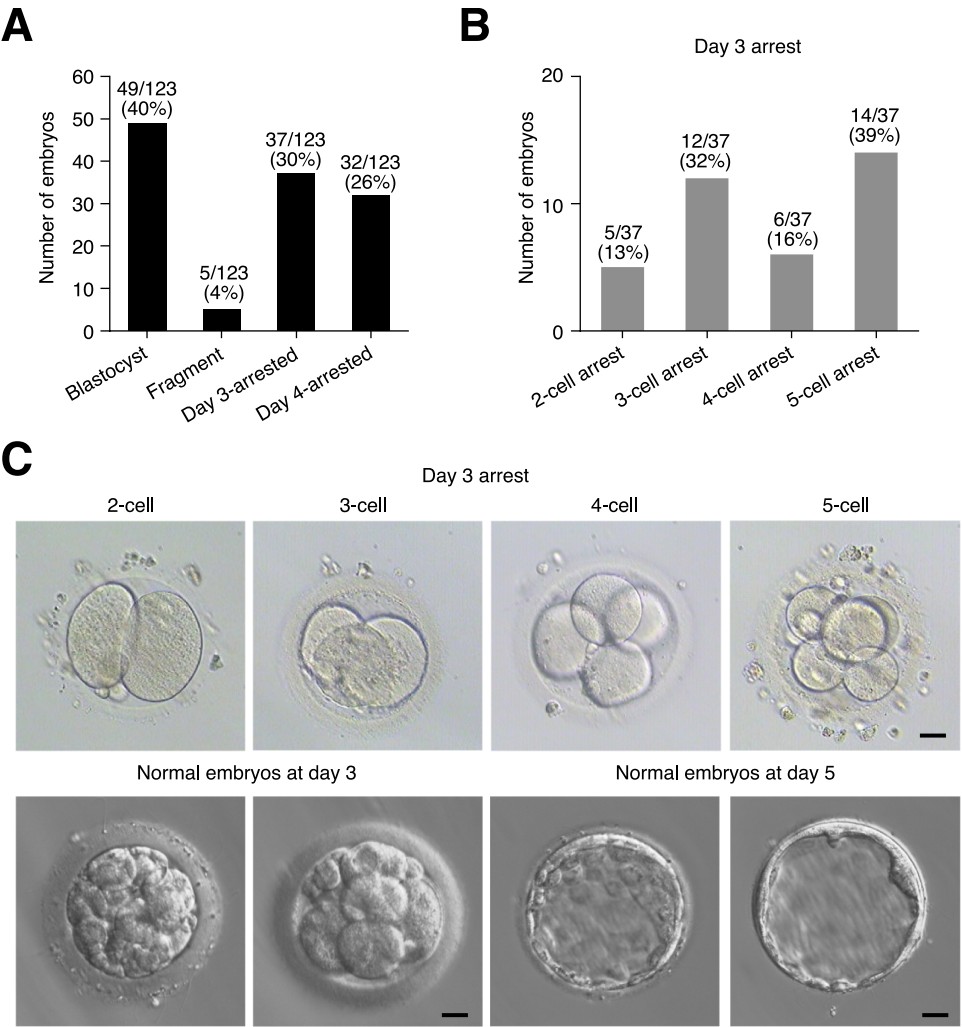

**Fig 1. Human IVF embryos have poor in vitro developmental capability.** (**A**) Typical outcomes for human IVF blastocyst development. Developmental results of 123 embryos were collected from 30 patients undergoing IVF. Underlying data can be found in **S1 Data**. (**B**) Number of blastomeres in the 37 embryos arrested on day 3 postfertilization. These embryos were defined as arrested and had normal morphology with distinct blastomeres, no unequal divisions, and no fragmentation. Underlying data can be found in **S1 Data**. (**C**) Bright-field images of uncompacted arrested human embryos between the 2-cell and 5-cell stages, and normal embryos at the same day of development. Scale bar = 20 μm. IVF, in vitro fertilized.

competency. For this study, we defined irreversibly arrested as day 3 embryos at the 2-cell to 5-cell stage that remained in that state on day 4, at which point a normal human embryo would be a morula (**Fig 1C**).

To explore the mechanism of arrest, we performed single-embryo RNA-seq on 17 arrested human embryos and combined this data with 6 arrested embryos from a previous study [4] (https://figshare.com/articles/dataset/Human_embryo_normalized_gene_expression_data/19775992). We compared these data with publicly available single-embryo or single-cell RNA-seq data from normal human embryos from the oocyte through to the late blastocyst [17–19]. Considering the high rate of developmental arrest of human embryos, it should be noted that the "normal" dataset is likely to contain embryos that are arrested or will go on to arrest. The dataset was analyzed using a hybrid single-cell RNA-seq and bulk RNA-seq pipeline based on

scTE and EDASeq G/C normalization [20–22]. In total, our dataset contained 1,020 single cells or single embryos, of which 23 were arrested.

## Arrested embryos can be divided into 3 types of arrest

We next investigated the developmental state of the arrested embryos. Potentially, arrested embryos have a failed or distorted developmental program, which may explain their arrest. Projection of the gene expression into principal component analysis (PCA) placed the arrested embryos in several locations, ranging from a 2-cell, 4-cell state, to 8-cell through morula (**Fig 2A**). This was surprising, as morphologically the cells remained as 2-cells to 5-cells (**Fig 1C**). The arrested embryos did not diverge from a normal developmental pathway and clustered with zygote through to the late morula (E4) stage. None of the arrested embryos had undergone compaction, yet many arrested embryos had a gene expression signature in advance of their morphological state. This suggests the developmental program is not correlated with the number of cells.

Early embryonic development is a highly dynamic process with rapid changes in gene expression [23,24]. The arrested embryos are distributed through several developmental stages, hence to isolate the factors involved in arrest, we attempted to remove development as a confounding variable. Co-correlation of the arrested embryos resulted in 3 groups (**Fig 2B**), which we designate Types I to III. Type I embryos clustered with the zygote, 2-cell and 4-cell stages, while Types II and III were grouped with 8-cell and beyond stages, (**Fig 2A**). Analysis of the arrested embryo types using CytoTRACE, which estimates developmental trajectories [25], placed Type I closest to the 4-cell stage, Type II between 4-cell and 8-cell, and Type III between E3 (embryonic-stage 3, early morula, as defined in [19]) and morula stages (**S1A–S1C Fig** and https://figshare.com/articles/dataset/Human_embryo_normalized_gene_expression_data/19775992). Interestingly, CytoTRACE did not indicate that the arrested embryos had lost developmental potency, supporting the idea that the developmental gene expression program is not substantially impacted by arrest. Analysis of genes specific to each stage of development indicated that each arrested type expressed genes normal for their developmental stage (**S1D–S1F Fig**). For example, Type I arrested embryos expressed typical developmental markers for the 4/8-cell stages, such as *LIN28A*, *DIS3*, *REST*, and *SNAPC1* (**S1D Fig**). While Types II and III arrested embryos expressed markers specific for the morula or even blastocyst, including *NANOG*, *DNMT3L*, *ESRRB*, and *ZFP42* (**S1E Fig**). Clustering the expression of genes from an 8-cell-specific signature identified in [26] clustered Type I with 2/4-cell stages, although some 8-cell-specific genes were up-regulated in the Type I embryos (**S1F Fig**). This suggests that the Type I arrested embryos are developmentally arrested at the 4-cell stage, but express some 8-cell markers, while Types II and III embryos are more developmentally advanced and express 8-cell, morula, and even blastocyst-stage developmental genes.

## Arrested embryos do not show excessive aneuploidy

We next looked at the karyotype of the arrested embryos. Normal human embryos have surprisingly high levels of aneuploidy, compared to other species [27], and this may be a contributing factor to arrest during IVF, particularly in embryos from women of advanced maternal age [28]. Aneuploidy mainly takes 2 forms [29]: full embryo aneuploidy due to meiotic errors in the oocyte/zygote or mosaic aneuploidy due to mitotic errors after fertilization [3,30]. We used the RNA-seq data to estimate the karyotype, as we could then correlate it with the arrest type in the same embryo. We used the method described in [31] to estimate aneuploidy (**S2 Data**). Around 30% of cells were predicted to be aneuploid (**S2A** and **S2B Fig** and **S2 Data**), which agrees with previous data that aneuploidy is common in human embryos [32]. We

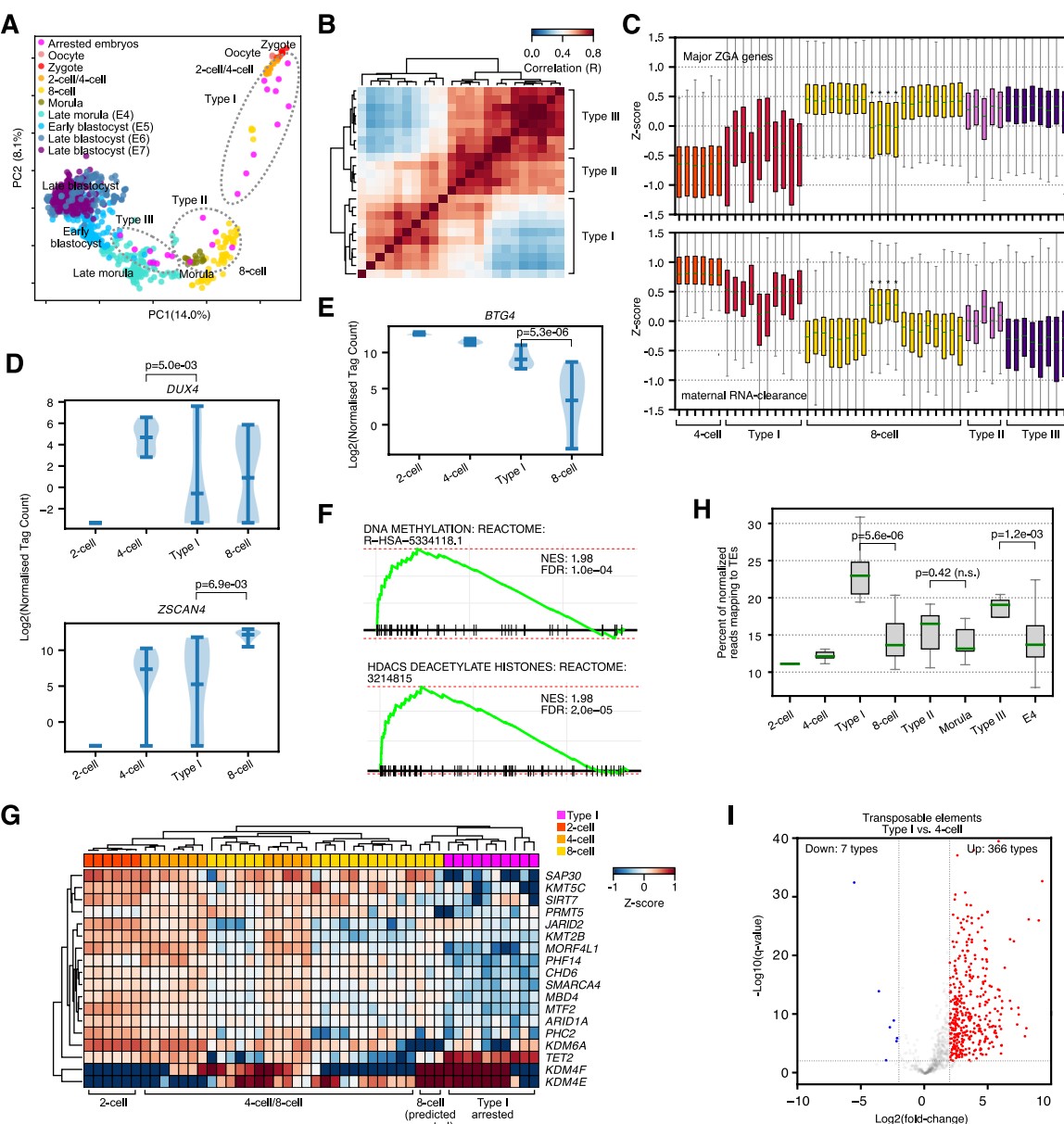

**Fig 2. Arrested embryos adopt 3 distinct types of arrest.** (**A**) PCA of single embryo or single-cell RNA-seq for normal or arrested human embryos. Arrested embryos are marked in pink. Presumed normal embryonic cells are colored by their developmental stage. The control (non-arrested presumed normal) samples are from a reanalysis of GSE66507 [18], PRJEB11202 [19], and GSE36552 [17] embryonic data. E = Embryonic-stage samples, as defined in [19], for this and all subsequent figures. Underlying data can be found in: https://figshare.com/articles/dataset/Human_embryo_normalized_gene_expression_data/19775992. (**B**) Pair-wise co-correlation matrix (Pearson's R) of the arrested embryo RNA-seq data. The types (based on the major clades in each cluster) are indicated. Clustering is based on Euclidean distance with complete linkage and optimal ordering. Underlying data can be found in: https://figshare.com/articles/dataset/Human_embryo_normalized_gene_expression_data/19775992. (**C**) Box plots showing the expression in each cell/embryo for the major ZGA genes and maternal RNA clearance genes as defined in **S3A Fig** and **S3 Data**. Cell types are labeled in the colored header bar and labeled below the heatmap. Note that the "normal" 8-cell from the [17] dataset, which appears to be failing the MZT and we predict is arrested, is indicated with stars. Underlying data can be found in: https://figshare.com/articles/dataset/Human_embryo_normalized_gene_expression_data/19775992. (**D**) Violin plot showing the expression of the key ZGA genes *DUX4* and *ZSCAN4*. Significance is from a 2-sided Welch's *t* test. Underlying data can be found in: https://figshare.com/articles/dataset/Human_embryo_normalized_gene_expression_data/19775992. (**E**) Violin plot of expression for the key maternal RNA clearance gene *BTG4*. Significance is from a 2-sided Welch's *t* test. Underlying data can be found in: https://figshare.com/articles/dataset/Human_embryo_normalized_gene_expression_data/19775992. (**F**) GSEA showing significantly enriched gene set terms for the up-regulated genes in the Type I arrested embryos versus 8-cell-stage embryos. Underlying data can be found in **S3 Data**. (**G**) Heatmap of the Z-scores of the expression of a selection of differentially expressed epigenetic factors. Cells/embryos were clustered according by Euclidean distance and complete

linkage. Underlying data can be found in: https://figshare.com/articles/dataset/Human_embryo_normalized_gene_expression_data/19775992. (**H**) Box plots showing the percent of normalized tags mapping to TEs in the indicated embryonic stages or the arrested embryos. Significance is from a 2-sided Welch's *t* test. Underlying data can be found in **S1 Data**. n.s. = not significant. (**I**) Volcano plot of differential gene expression for Type I arrested embryos versus 4-cell-stage embryos. This plot only shows the differentially expressed TEs. The x-axis is the log2 fold-change, and the y-axis is the −log10(q-value) as reported by DESeq2 with Bonferroni–Hochberg multiple testing correction. Significantly, up- and down-regulated TEs are labeled in red and blue, respectively, and the number of TE types that are up- or down-regulated are labeled. Underlying data can be found in **S4 Data**. GSEA, gene set enrichment analysis; MZT, maternal-to-zygotic transition; PCA, principal component analysis; TE, transposable element; ZGA, zygotic genome activation.

notice that chromosomal defects become particularly evident at the 8-cell stage, and reach around 30% at the morula/E3 stage (14/48 cells, 29%) and persist to the E7-stage (late blastocyst) at similar rates (116/321 cells, 36%) (**S2C Fig**). Of the arrested embryos, 6/23 (26%) had a predicted aneuploidy (**S2A and S2B Fig**). This number is not substantially different from normal embryos and is in line with the typical levels of aneuploidy seen in human embryos. This computational approach cannot easily detect the difference between meiotic and mitotic aneuploidies, but assuming most of the aneuploidies we see are due to mitotic errors, there was no overall bias in the gain or loss of specific chromosomes (**S2C and S2D Fig**). Ultimately, these data suggest that aneuploidy is not a specific feature of arrested embryos. In a study of aneuploidy in embryos from women of advanced maternal age, 50% of embryos still developed to the blastocyst stage, despite 84% of the embryos having at least 1 chromosomal abnormality [28]. Similarly, there is evidence that mosaic aneuploidies are common and may not be detrimental to development [30,33], at least to the blastocyst stage. Finally, meiotic aneuploidies can develop to the blastocyst stage, although they have severe consequences for further development [34]. Hence, we argue that while aneuploidy is an important problem in postimplantation development, it is not responsible for developmental arrest pre-compaction or to reach the blastocyst.

## Type I arrested embryos fail the maternal-to-zygotic transition

We next looked at biological processes that were behind the arrest. Because each type of arrested embryo clusters with different embryonic stages, we decided to investigate each type separately. The arrest of Type I embryos is developmentally close to the maternal-to-zygotic transition (MZT), which happens from fertilization to the 8-cell stage [35]. The MZT encompasses 2 processes, major ZGA and the degradation of maternal transcripts. A failure of the major ZGA or maternal-clearance may lead to embryonic arrest. Indeed, there is evidence that maternal-clearance is defective in arrested embryos [4], but the major ZGA can initiate normally in arrested embryos [36]. We first defined major ZGA genes as those genes that were significantly up-regulated >4-fold from 2-cell to 8-cell stages, and maternal-clearance genes as those significantly down-regulated (**S3A Fig** and **S3 Data**), in a method similar to [37]. Gene set enrichment analysis (GSEA) supported the designation of these genes as representing the MZT, as up-regulated terms included spliceosomes, transcription, and down-regulated genes included female gamete generation (**S3B Fig**).

We next applied these 2 MZT gene sets to the arrested embryos. Surprisingly, the expression of major ZGA genes and maternal-clearance genes could discriminate Type I arrested embryos from Types II and III (**Fig 2C**). Types II and III arrested embryos had gene expression levels of major ZGA genes that matched the 8-cell stage, and maternal-clearance genes were lower in Type II/III than in 4-cell-stage embryos (**Fig 2C**), indicating that Types II and III arrested embryos had traversed the MZT. Conversely, the Type I arrested embryos had poor activation of ZGA genes and incomplete degradation/reduction of maternal transcripts (**Fig 2C**). This observation was supported by the expression of key MZT-related regulatory genes.

The DUX family of transcription factors (TFs) are involved in ZGA, and *DUX4* is activated just before major ZGA [38,39]. In the Type I arrested embryos, *DUX4* was low compared to the expression of *DUX4* in 4-cell-stage embryos, and 2 target genes of DUX4, *DUXA*, and *ZSCAN4* were poorly induced compared to the 8-cell-stage (**Figs 2D** and **S3C**). Similarly, the key maternal RNA-clearance genes *BTG4*, *PAN2*, and *CNOT6L* [40] remained high in Type I arrested embryos (**Figs 2E** and **S3D**). Interestingly, our data indicates that one of the "normal" 8-cell embryos resembles a Type I arrested embryo, with low levels of major ZGA genes and incomplete maternal RNA-degradation (**Fig 2C**). Overall, our data suggest that MZT failure can account for about approximately 40% of arrested embryos (10/23 arrested embryos and 1/4 "normal" embryos).

We next looked at mechanisms underlying the arrest of Type I embryos. The MZT is a time of very active epigenetic and 3D genome rearrangements [24], and we speculated that epigenetic defects underlie MZT. We first measured differentially expressed (DE) genes and transposable elements (TEs) by comparing Type I versus 4-cell-stage embryos (**S4A Fig**). GSEA indicated that DNA methylation and HDAC deacetylase pathways were up-regulated (**Fig 2F**), suggesting epigenetic regulatory dysfunction. Many specific epigenetic repressors and activators were significantly down-regulated in arrested embryos (**Fig 2G**). For example, *SAP30*, a member of the SIN3A co-repressor complex was down-regulated, along with the histone arginine methyltransferase *PRMT5*. Histone lysine methylating enzymes were also down-regulated, including *KMT5C (SUV420H2)*, which is responsible for catalyzing the repressive histone H4K20me3 mark. Additionally, *KMT2B (MLL2)*, an enzyme responsible for the active histone mark H3K4me3 was also down-regulated. Conversely, the histone demethylases *KDM4F* and *KDM4E* were up-regulated in arrested embryos (**Fig 2G**). The consequences of these changes are likely to be major disruptions in the balance of activatory and repressive chromatin.

Ideally, we would perform ChIP-seq for methylated histones in human embryos. However, this technique is extremely challenging when small amounts of material are available. Histone ChIP-seq has been performed using mouse embryos but required several hundred embryos [41], which is a level of material not available for human embryos. Consequently, we exploited an indirect method to measure the epigenetic state of the cell. In other embryonic model systems, such as mouse or human ESCs, we and others have shown that when epigenetic regulators are disrupted, TEs tend to be up-regulated [42,43]. The situation is complicated by the presence of expressed TE sequences during normal embryonic development [44]. Nonetheless, TE expression can act as an indirect read-out for chromatin state. Remarkably, in the Type I arrested embryos, we observed a dramatic increase in the overall number of reads mapping to TEs (**Fig 2H**). This was also reflected in the number of differentially regulated TEs, and 366 TE types were up-regulated, while only 7 types of TE were down-regulated (**Fig 2I**). This pattern held whether we compared Type I to 2-cell, 4-cell, or 8-cell cells (**S4B Fig**). TEs were unaffected in Type II arrested embryos, and modestly up-regulated in Type III (**Fig 2H**); however, only a few types of TE were differentially expressed (**S4C Fig**). In Type I arrested embryos, endogenous retrovirus (ERV) families were up-regulated, and approximately 70% of ERV1, ERVL, and ERVK family TEs were up-regulated (**S4D Fig**). Only a few SINE TEs were up-regulated, but LINE:L1s were up-regulated, including the L1HS family of TEs that are capable of transposition [45] (**S4E Fig**). Indeed, a major epigenetic suppressor of LINE L1s, PRMT5 [46], was significantly down-regulated in Type I arrested embryos (**Fig 2G**). Ultimately, the massive deregulation of TEs seen in arrested embryos is reminiscent of the widespread activation of TE families in response to the knockdown of epigenetic regulators we previously observed in mouse ESCs [42]. One further point to note is that many of the deregulated epigenetic factors in Type I embryos are repressors (**Fig 2G**). This may seem counterintuitive that repressors are

down-regulated, TEs are up-regulated, but the ZGA fails to activate. However, epigenetic repression is an important, if incompletely understood, component of the ZGA [47]. Overall, our data suggest that Type I arrested embryos have MZT defects that are likely due to epigenetic deregulation.

## Types II and III arrested embryos enter a senescent-like state

We next looked at Types II and III arrested embryos. These arrested embryos are distinct from Type I, by overall gene expression patterns (**Fig 2B**), successful traversal of the MZT (**Fig 2C**), and do not appear to have a deregulated epigenetic state, as measured by TE sequence fragment expression (although LINE-1 L1HS are up-regulated) (**S4C** and **S4E Fig**). This suggests other mechanisms are responsible for Types II and III arrest. Embryonic development is a dynamic process, PCA and CytoTRACE indicates the Types II and III arrested embryos are most similar to cells of the 8-cell/morula stage and the late morula (E4)-stages, respectively (**Figs 2A** and **S1B**). To determine the closest comparable embryonic stage, we measured the number of significantly DE genes between Types II and III embryos versus 8-cell, morula, and E4-stages (**S5A Fig**), and chose the comparison with the smallest overall number of DE genes. This approach suggested that the closest comparisons are Type II versus morula (1784 DE genes) and Type III versus E4 (824 DE genes) (**S5A and S5B Fig** and **S3 Data**).

We next analyzed the sets of genes that were DE for Types II and III. Interestingly, for Types I and II, ribosomes, histones, and translation-related genes were down-regulated, as determined by GSEA and gene expression levels (**Fig 3A–3C** and **S4 Data**). In Type III embryos, large ribosomes were significantly down-regulated, while small ribosomes were unaffected and nucleosomes were significantly elevated; however, select histones and ribosome transcripts were significantly down-regulated in all 3 arrested embryo types (**S5C and S5D Fig**). Ranking cells by the sum of expression of small and large ribosomes and nucleosomes placed almost all arrested embryos at the bottom of the list (**S5E–S5G Fig**). Based on the down-regulation of nucleosomes and ribosomes, we reasoned that the arrested embryos were entering into a senescent-like state. This was supported by GSEA, which suggested a senescence-like gene expression program was being activated (**Fig 3D**). Cell cycle genes were deregulated in the arrested embryos. Expression of the A-type cyclin *CCNA2* was down-regulated, while the cell cycle inhibitor *CDKN1A* (p21) was up-regulated in arrested embryos (**Fig 3E**). Marker genes specific to cell cycle phases tended to be lower in the arrested embryos [48] (**S6A and S6B Fig**). These included genes involved in an active cell cycle, such as the tubulin subunits *TUBB4B*, *TUBA1A*, *CCNB1*, and *PCNA*, which were significantly down-regulated in arrested embryos (**S6A and S6B Fig**). These results suggest that the arrested embryos are entering into a senescent-like state marked by reductions in ribosomes, nucleosomes, protein translation, and cell cycle factors.

Senescence and quiescence are defined molecular programs with overlapping signatures [49–51]. A key regulator of senescence and quiescence in hematopoietic stem cells (HSCs) is p53 [52]. GSEA of up- and down-regulated genes suggested hyperactivation of p53 target genes (**Fig 3F**). The mRNA levels of p53 (*TP53*) were either unaffected (Type II) or slightly reduced (Type III) in the arrested embryos (**S6C Fig**). However, immunofluorescence indicated that arrested embryos had high levels of phosphorylated (Ser15) p53, a mark of activation (**Fig 3G**). A major target gene of p53, *MDM2*, was also significantly up-regulated (**Fig 3H**), a feature also seen in quiescent HSCs [52]. Finally, ranking all cells by the sum of their gene expression for p53 target genes placed all arrested embryos at the top of the list (**S6D Fig**), indicating increased activity downstream of p53. This analysis suggests that the arrested embryos are becoming senescent, with a deregulated cell cycle and activated p53. However, the

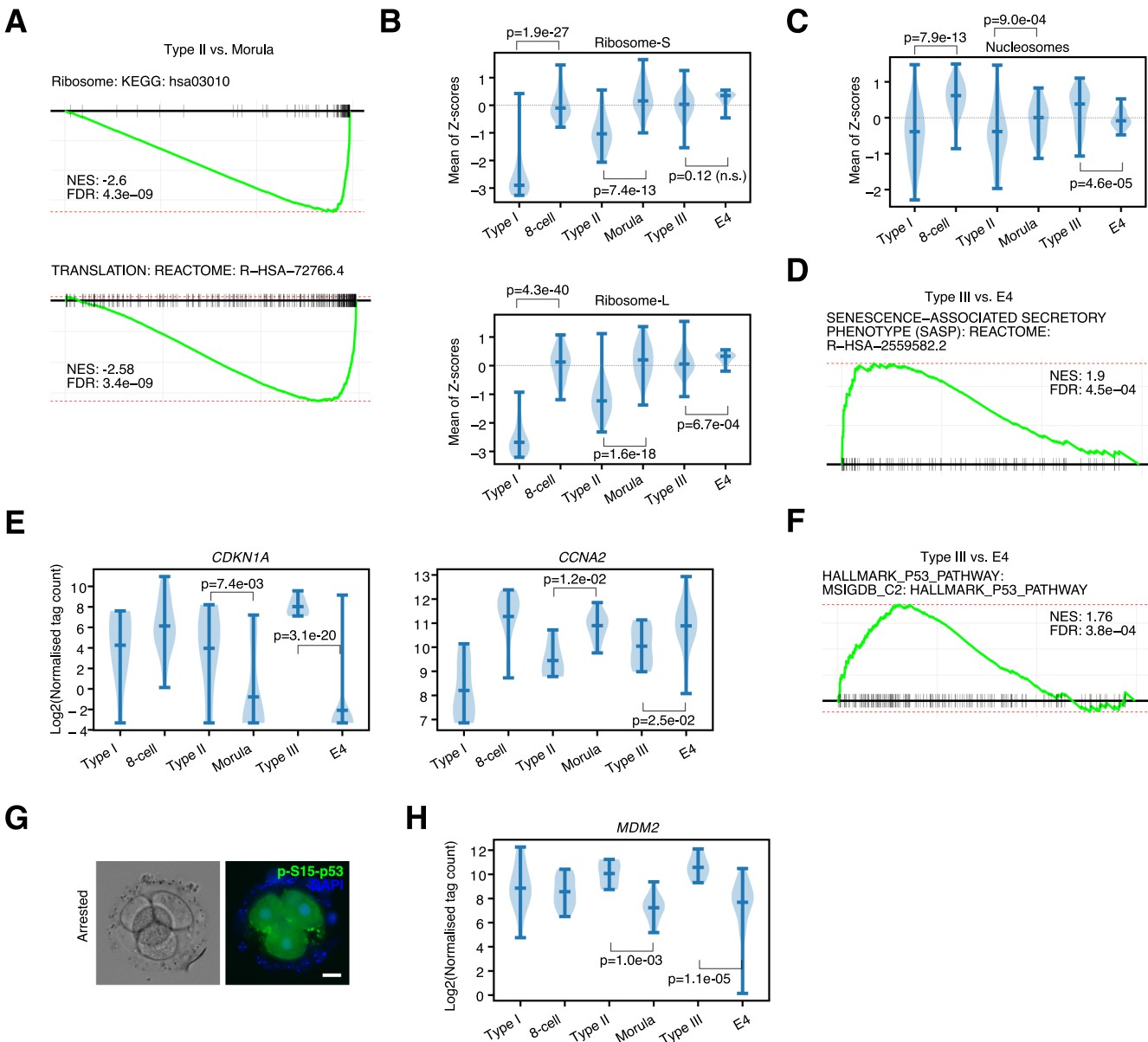

**Fig 3. Arrested embryos down-regulate histones, ribosomes, and adopt a senescent-like gene expression program with elevated p53 activity.** (**A**) GSEA for differentially expressed genes in the Type II versus morula comparison. Underlying data can be found in **S5 Data**. (**B**) Violin plot of the means of the Z-scores for all small (top) and large (bottom) ribosomal subunits in the indicated embryonic stages or arrested embryos. Significance is from a 2-sided Welch's *t* test. Underlying data can be found in **S1 Data**. n.s. = not significant. (**C**) Violin plot of the means of the Z-scores for all expressed nucleosomes in the indicated embryonic stages or arrested embryos. Significance is from a 2-sided Welch's *t* test. Underlying data can be found in **S1 Data**. (**D**) GSEA showing the SASP term is significantly enriched in Type III versus E4 (late morula) comparison. Underlying data can be found in **S5 Data**. (**E**) Violin plot showing the expression of *CDKN1A* (p21), or *CCNA2* in the indicated embryonic stages or arrested embryos. Significance is from a 2-sided Welch's *t* test. Underlying data can be found in: https://figshare.com/articles/dataset/Human_embryo_normalized_gene_expression_data/19775992. (**F**) GSEA showing the enrichment of the HALLMARK p53 list of target genes is significantly enriched in Type III versus E4 (late morula) comparison. Underlying data can be found in **S5 Data**. (**G**) Phospho-S15-p53 (green) immunostaining in arrested embryos. Embryos are co-stained with DAPI (blue), scale bar = 20 μm. (**H**) Violin plot of expression showing the expression level of the p53 target gene *MDM2* in the indicated embryonic stages or arrested embryos. Significance is from a 2-sided Welch's *t* test. Underlying data can be found in: https://figshare.com/articles/dataset/Human_embryo_normalized_gene_expression_data/19775992. GSEA, gene set enrichment analysis; SASP, senescence-associated secretory phenotype.

developmental program is still being executed, which suggests that development, senescence, and cell cycle are uncoupled processes.

## Partial rescue of arrested embryos by resveratrol

The arrested embryos have a disrupted cell cycle. However, it is unclear if the arrest is related to quiescence (i.e., reversible) or senescence (i.e., irreversible). Senescence and quiescence have many similar cellular and mechanistic features, and there is no clear way to discriminate between the 2 states, except for reversible reactivation [50]. To determine if the arrested embryos are irreversibly arrested, we attempted to reactivate the embryos and recommence development. We selected several small molecule inhibitors that have previously been shown to impact embryonic or pluripotent stem cell development. Specifically, the mTOR inhibitor rapamycin, ERK inhibitor PD0325901, vitamin C, and resveratrol. These 4 compounds have been implicated in various aspects of senescence and embryogenesis. Rapamycin inhibits mTOR to promote autophagy and has been shown to improve pig oocyte development [53]. Vitamin C is an antioxidant and epigenetic modulator that can affect DNA methylation by functioning as a co-factor for DNA demethylation TET enzymes [54]. ERK inhibition assists in the establishment of the inner cell mass in the blastocyst stage [55]. Finally, resveratrol is an antioxidant that can improve pig and bovine oocytes [56–58] and in vitro development of aged mice and human oocytes [59].

Application of rapamycin, PD0325901, and vitamin C to the arrested embryos had only a limited impact, and the embryos did not recommence development at rates substantially higher than control (untreated arrested) embryos (S7A Fig). Only resveratrol, an antioxidant with SIRT-activating effect, had a substantial impact on the development of arrested embryos. After treatment, 23/42 embryos recommenced development (Fig 4A). However, it should be noted that 4/42 embryos reinitiated development, but ultimately fragmented, suggesting that at least some of the reactivated embryos are incapable of further development. Similarly, while many (19/42, 45%) of the embryos recommenced development, only 9 compacted, and only 3 made it to the blastocyst stage (Fig 4B and 4C). A caveat should, however, be applied. The arrested embryos were cultured for a further day before starting treatment with resveratrol (i.e., the day 3–arrested embryos started treatment on day 4). Potentially, treating the embryos with resveratrol at an earlier time point may have activated more embryos without disintegration.

To further investigate the gene expression patterns, we performed single-embryo RNA-seq on the resveratrol-treated embryos that recommenced development and made it to at least the morula stage. PCA suggested that the resveratrol-treated embryos had indeed recommenced development (Fig 4D), as the resveratrol-treated embryos were now grouped with early and late blastocyst-stage embryos, and normal developmental marker genes were activated. For example, *ESRRB* and *TFCP2L1* were highly expressed (S7B Fig). TEs were also expressed at normal levels (S7E and S7F Fig). We next looked at senescent and cell cycle–related genes. Curiously, ribosomes and nucleosomes had not recovered to their normal levels (S7C Fig) nor had the cell cycle–related genes *CENPA* and *CCNA2* (S7D Fig). Expression of the cell cycle inhibitor *CDKN1A* had also not declined (Fig 4E). However, immunostaining of arrested embryos and embryos treated with resveratrol indicated that the protein level of p21 (*CDKN1A*) was reduced, suggesting resveratrol is affecting p21 posttranslationally (Fig 4F). Similarly, phosphorylated AKT (Ser473) was also reduced (Fig 4G), suggesting reduced cross-talk between RB and AKT, which is a hallmark for reduced senescence in mouse liver cells [60]. Overall, resveratrol partially reactivated a normal developmental program in a minority of embryos. However, the resveratrol-treated embryos retained deregulated levels of ribosomes, protein translation, and have not adopted a fully normal state.

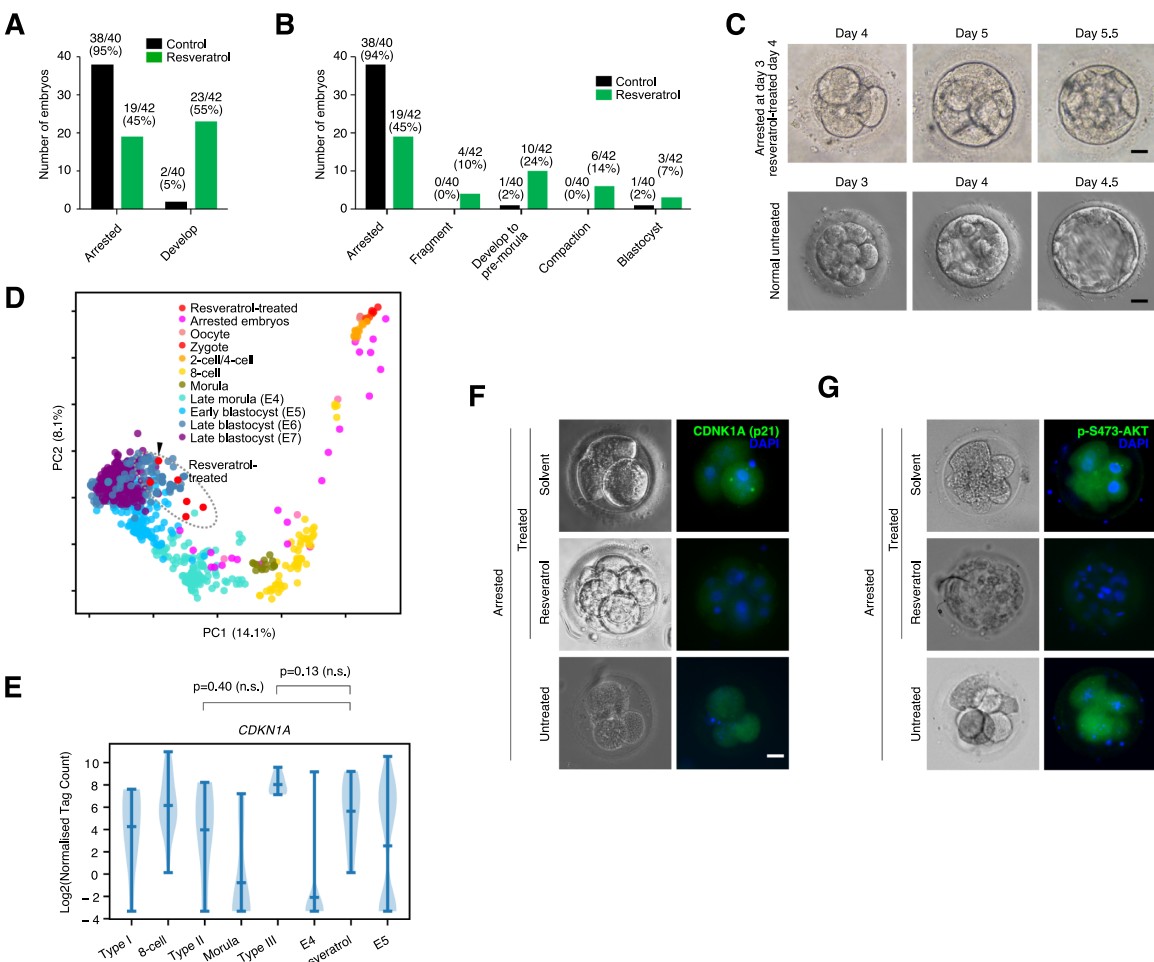

**Fig 4. Resveratrol can overcome embryonic arrest in limited cases.** (**A**) Number of embryos that remain in an arrested state or recommenced development (including those embryos that fragment). Embryos were either left untreated or treated with resveratrol. The total number of embryos in the control group is 40, and 42 in the resveratrol group. Underlying data can be found in **S1 Data**. (**B**) Bar chart showing the breakdown of the stages of embryonic development the control and resveratrol embryos reached. Underlying data can be found in **S1 Data**. (**C**) Morphology of a reactivated resveratrol-treated embryo, compared to normal untreated embryos. The embryo was arrested on day 3 and did not show any degeneration at day 4. After treatment with resveratrol, the embryo proceeded to the early blastocyst-like stage. Scale bar = 20 μm. (**D**) PCA of normal human embryo RNA-seq, with arrested embryos (pink) and embryos treated with resveratrol (red). The embryo in **panel C** is marked with a black arrow. Underlying data can be found in: https://figshare.com/articles/dataset/Human_embryo_normalized_gene_expression_data/19775992. (**E**) Violin plot showing the expression of *CDKN1A* in the indicated embryonic stages, arrested embryos, or the reactivated embryos treated with resveratrol. Underlying data can be found in: https://figshare.com/articles/dataset/Human_embryo_normalized_gene_expression_data/19775992. n.s. = not significant. (**F**) Immunostaining and brightfield of embryos either untreated or treated with solvent or with resveratrol. Immunofluorescence using an antibody against p21 (*CDKN1A*) (green). Embryos are co-stained with DAPI (blue), scale bar = 20 μm. (**G**) As in **panel F**, but for phospho-S473-AKT (green). Embryos were co-stained with DAPI (blue), scale bar = 20 μm. PCA, principal component analysis.

## SIRT activators resveratrol and nicotinamide riboside affect embryonic metabolism

Resveratrol has 2 main mechanistic functions: as an antioxidant and also as a co-activator of the NAD+ dependent deacetyltransferase SIRT1 [61]. RNA-seq analyses showed that the expression *SIRT1* mRNA was low in Type I arrested embryos, but was high in Type II/III and resveratrol-treated arrested embryos (**Fig 5A**). Using immunofluorescence, SIRT1 protein levels were low in the arrested embryos, but when treated with resveratrol, SIRT1 levels increased

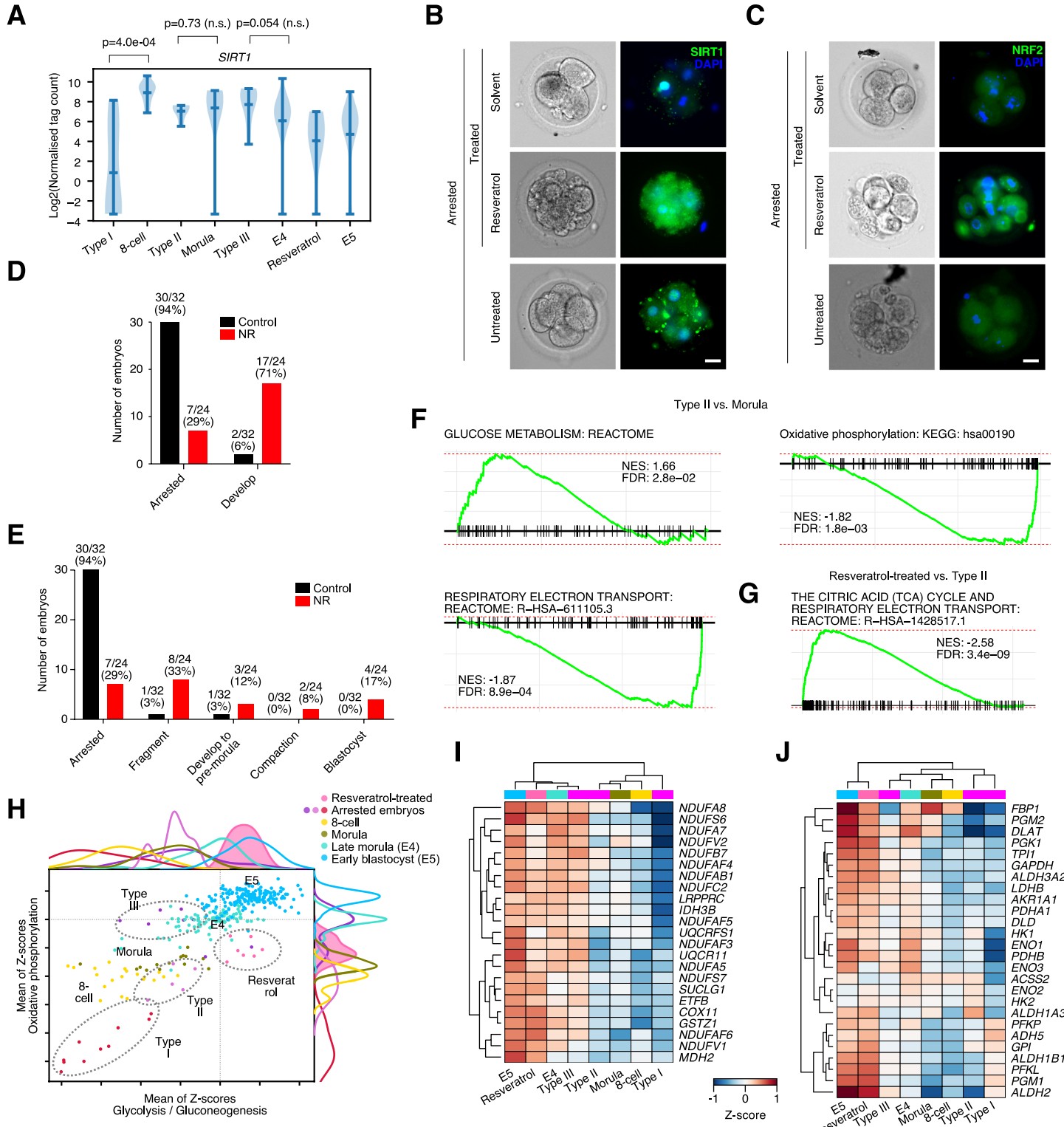

**Fig 5. Resveratrol and nicotinamide riboside reactivate development partly through modulation of SIRT activity. (A)** Violin plot showing the expression of *SIRT1* in the indicated embryonic stages or arrested embryos or resveratrol-reactivated embryos. Underlying data can be found in: https://figshare.com/articles/dataset/Human_embryo_normalized_gene_expression_data/19775992. n.s. = not significant. **(B)** Immunofluorescence staining of SIRT1 (green) and brightfield in arrested untreated of treated with solvent or resveratrol-reactivated embryos. Embryos are co-stained with DAPI (blue), scale bar = 20 μm. **(C)** Immunofluorescence staining of NRF2 (green) and brightfield in arrested untreated or treated with solvent or resveratrol-reactivated embryos. Embryos were co-stained with DAPI (blue), scale bar = 20 μm. **(D)**

Number of embryos that remain in an arrested state or recommenced development (including those embryos that fragment). Embryos were either left untreated or treated with NR. The total number of embryos in the control group is 32 and 24 in the NR-treated group. Underlying data can be found in S1 Data. (E) Bar chart showing the number of arrested embryos that would reinitiate development (including embryos that would ultimately fragment). The total number of embryos in the control group is 32 and 24 in the NR-treated group. Underlying data can be found in S1 Data. (F) GSEA for up- and down-regulated genes in Type II versus morula comparison. Underlying data can be found in S5 Data. (G) GSEA for up- and down-regulated genes in resveratrol versus Type II arrested embryos. Underlying data can be found in S5 Data. (H) 2D dot plot showing the sum of the Z-scores for the genes in the indicated KEGG categories. The x-axis scores the glycolysis/gluconeogenesis pathway, the y-axis the oxidative phosphorylation pathway. Each dot in the plot is a cell/embryo, and the top and right axis have the kernel density for each group of cells. The resveratrol-treated embryos have a filled-in color (pink) for emphasis. The locations of the arrested and resveratrol-treated embryos are indicated by dashed lines and labels, and the normal developmental states are indicated by labels. Underlying data can be found in S1 Data. (I) Heatmap showing the expression of select oxidative phosphorylation genes. Underlying data can be found in S1 Data. (J) Heatmap showing the expression of select glycolytic metabolic genes. Underlying data can be found in S1 Data. GSEA, gene set enrichment analysis; NR, nicotinamide riboside.

(Fig 5B). We also measured NRF2 protein levels (Fig 5C), as resveratrol has been reported to confer its antioxidant benefits by up-regulating NRF2 at both the transcriptional and protein levels [62]. Resveratrol indeed activated the antioxidant protein NRF2. These data suggest that resveratrol is activating both SIRT and an antioxidant effect, although it is not clear which of the 2 are important for the reactivation of the arrested embryos.

The antioxidant vitamin C failed to reactivate arrested embryos (S7A Fig), which suggests an antioxidant effect is not the main factor for embryo reactivation. Hence, resveratrol may be reactivating development by modulating SIRTs. We next treated arrested embryos with a second SIRT activator NR that does not have reported antioxidant capability. NAD+ availability is a rate-limiting step for SIRT activity, and SIRTs can be activated by NR that is converted to NAD+ and increases the activity of SIRT1 and SIRT3 in cells [63]. When arrested embryos were treated with NR, in an effect similar to resveratrol, they were reactivated and would proceed through development and a few embryos reached a blastocyst-like state (Fig 5D and 5E). As NR and resveratrol can phenocopy, it suggests that the antioxidant activity of resveratrol is not required, and the dominant pathway in the reactivation of arrested embryos is the activation of SIRTs.

SIRT enzymes have a key role in controlling the balance of glycolytic, oxidative, and fatty acid metabolic processes in somatic cells [63,64]. The developing human embryo undergoes significant changes in the metabolic pathways utilized at each embryonic stage. However, these metabolic changes remain somewhat unclear due to the difficulty of directly assaying metabolic products in small numbers of cells [65]. Briefly, from the zygote to the morula, embryos use a form of oxidative metabolism based on pyruvate. In the preimplantation blastocyst, embryos convert to a balanced glycolytic/oxidative phosphorylation-based metabolism using glucose as a fuel source, before transitioning to glycolysis in the low oxygen environment after implantation [65]. Studies in human and mouse naïve and primed PSCs, which resemble the preimplantation and the postimplantation epiblast, respectively [66], suggest that HIF1A and the balance between SIRT1 and SIRT2 activity is important [67,68].

To explore metabolism in the arrested embryos, we would ideally measure the metabolic products directly. However, this is infeasible with current technology, which requires hundreds or thousands of embryos for mass spectrometry-based approaches [69,70]. Hence, we attempted to infer metabolic state based upon RNA-seq data. This approach is fraught with difficulty as metabolic enzymes generally have high steady-state levels of mRNA that do not respond to changes in metabolism. Consequently, we infer metabolism based on the changes in RNA levels of sets of metabolic-associated transcripts. GSEA indeed suggested that glucose metabolism was up-regulated and oxidative phosphorylation was down-regulated in Type II arrested embryos (Fig 5F). GSEA for resveratrol-treated embryos also supported increased expression of oxidative phosphorylation genes when we compared the resveratrol-treated embryos to Types II or III arrested embryos (Figs 5G, 5H, S8A, and S8B). We employed a

technique that used the sum of Z-scores for a set of genes to infer pathway activity. Plotting the sum of Z-scores of oxidative phosphorylation versus glycolysis/gluconeogenesis gene sets suggested that resveratrol was pushing cells toward increased glycolytic metabolism, whereas all the arrested cells had reduced glycolysis (and variable levels of oxidative phosphorylation) (**Fig 5H**). This is illustrated by the RNA levels of oxidative phosphorylation and glycolysis genes that were higher in resveratrol-treated embryos (**Figs 5I, 5J,** and **S8C**). It should be noted though that resveratrol had not returned the reactivated embryos to a completely normal expression state, and there remain several hundred genes that are significantly differentially expressed between resveratrol-treated and E5 (early blastocyst) embryo cells, including many biological pathways that remain low (**S8D and S8E Fig**). Similarly, overall pattern of metabolism had not reached the same state as normal E4 and E5 embryos (**Fig 5H**).

Finally, in our analysis of metabolism, we noticed that resveratrol also up-regulated fatty acid metabolism-related genes (**S8B**, **S9A, and S9B** **Figs**). This is reminiscent of the situation in mouse hepatocyte cells, where the loss of *Sirt1* leads to reduced fatty acid oxidation, due to deregulation of PPAR-genes [71], and agrees with several studies that show resveratrol activates SIRT1 to decrease lipogenesis and increase fatty acid oxidation in many cellular contexts [72]. Overall, our data suggest that arrested embryos erroneously maintain an oxidative phosphorylation-biased metabolism and low levels of fatty acid oxidation and glycolysis (**S9C Fig**).

## Inferred transcriptional regulation in arrested human embryos

We next explored transcriptional regulation in the arrested embryos. The direct assay of TF binding to the genome is currently impractical in human embryos. To date, TF binding in single cells has only been mapped using a system that involves transposons to introduce novel insertions that can then be recovered from RNA-seq data [73]. This technique requires transgene transfection and is thus impractical in arrested embryos. Chromatin accessibility has been performed in single cells [74]; however, for an individual cell, the data remain extremely sparse and chromatin accessibility binding is determined by pooling data from relatively large numbers of similar single cells to reconstruct chromatin accessibility. Hence, we reverted to an approach that takes advantage of the fact that TF binding is rich around the transcription start sites (TSSs) of DE genes [75,76]. In total, we found many TF motifs enriched in the DE genes in each type of arrest (**S10A Fig**). To understand the pattern of transcriptional regulation, we focused on TF families known to regulate quiescence/senescence and metabolism. The TF FOXO1 is a major regulator of senescence in somatic cells and works partly by down-regulating MYC activity [77]. *FOXO1* mRNA varied, and was significantly down-regulated in Type I embryos, unchanged in Type II and significantly up-regulated in Type III (**S10B Fig**). However, the FOXO1 TF-binding motif was significantly enriched in the promoters of all 3 types of arrested embryos (**Fig 6A**). This suggests that while the mRNA levels of *FOXO1* vary, FOX-family activity is increased in the arrested embryos. The MYC motif was enriched in Types I and III DE genes, and MYC target genes were down-regulated in Types II and III arrested embryos (**Fig 6B**). This agrees with the observation that MYC activity is low in diapause and senescent mouse blastocysts [78]. High levels of RUNX-family TF activity has been linked with senescence in stem cells [79], and a RUNX-family motif was enriched in all DE genes (**Fig 6A**). Overall, this TF-binding inference suggests that 3 TFs implicated in senescence, MYC, FOX, and RUNX families, are active in the arrested embryos.

Finally, we looked at the transcriptional and signaling pathways in the resveratrol-treated arrested embryos versus the Type III arrested embryos (as the Type III arrested embryos are developmentally closest to resveratrol-treated embryos). Motif enrichment and GSEA in arrested embryos suggested the up-regulation of HIF-family TFs (**Fig 6A**), and the HIF

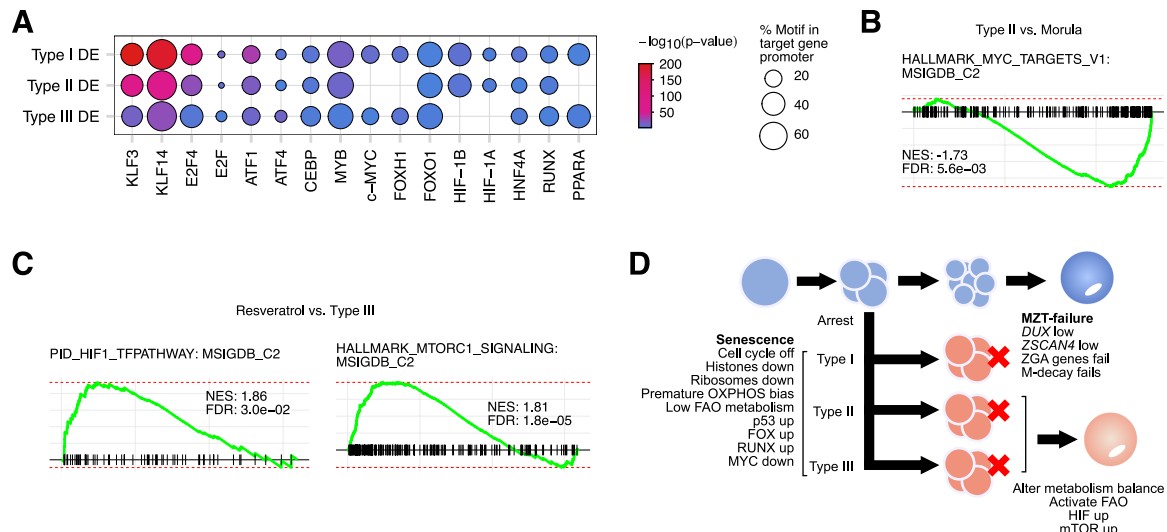

**Fig 6. Transcriptional control of embryonic arrest.** (**A**) Dot plot for enriched TF motifs in the DE genes in the indicated arrested embryo types. Motif discovery was performed using HOMER with default settings [98] against the promoters (defined here as −1,000 bp upstream of the TSS) of the DE genes in the indicated types of arrested embryo. The size of the circle indicates the percent of promoters that had the motif, and the color indicates the enrichment *p*-value. (**B**) GSEA of Type II versus morula, this panel shows the HALLMARK MYC-targets gene set. Underlying data can be found in **S5 Data**. (**C**) GSEA of resveratrol-treated embryos versus Type III arrested embryos. Underlying data can be found in **S5 Data**. (**D**) Schematic model of the types of arrest and the underlying pathways driving the arrest. Types I–III cells all enter a senescent-like state, driven by activated p53, and low MYC activity. Histones and ribosomes are down-regulated and cell cycle activity is reduced. Type I arrested embryos harbor an MZT developmental error. Types II and III embryos can be induced to recommence development by modulating SIRT activity. DE, differentially expressed; FAO, fatty acid oxidation; GSEA, gene set enrichment analysis; M-decay, maternal RNA decay; MZT, maternal-to-zygotic transition; OXPHOS, oxidate phosphorylation; TF, transcription factor; TSS, transcription start site; ZGA, zygotic genome activation.

pathway genes were up-regulated in resveratrol-treated embryos (**Fig 6C**). This agrees with a HIF1a-driven switch from oxidative phosphorylation-based metabolism to glycolytic, as previously seen in naïve and primed mouse and human PSCs [68]. This was supported by the enrichment of PPAR-family TF motifs in the arrested DE genes (**Fig 6A**), which is in agreement with PPAR-deregulation in *Sirt1* knockouts in mouse [71].

## Discussion

Human (and primate) embryos are surprisingly poor at developing in vitro when compared with other mammalian species. The poor developmental capacity of human embryos has major implications for IVF. Here, we show that the arrested embryos enter a senescent-like state. The arrested embryos down-regulated histones, ribosomes, and RNA processing machinery, and for TF activity p53, FOX, RUNX were up-regulated, while MYC activity was down-regulated. We classified the arrested embryos into 3 types based on gene expression (**Fig 6D**). Type I appear to have failed the MZT, while Types II and III appear to have metabolic problems.

There are similarities and differences between the 3 types of arrest, they all arrest and acquire a senescent-like molecular program. Type I fail to correctly regulate MZT genes, which, based on deregulation of TEs we speculate is due to problems in epigenetic regulation. The difference between Types II and III is less clear, although Type III has increased levels of genes related to oxidative phosphorylation, compared to Types I and II. Resveratrol and NR could partially overcome the arrest of the embryos; however, when we treated the arrested embryos, we did not know the type of arrest. This may help explain why resveratrol or NR can

activate only a subset of arrested embryos. We speculate that resveratrol or NR is primarily activating the Type II/III arrested embryos, by altering SIRT activity and metabolism. Conversely, we believe resveratrol or NR has little effect on Type I embryos, as they fail to complete the MZT.

Limitations should be highlighted in this study. Firstly, our access to normal embryos is extremely limited, and this limits some of the experiments performed. We were unable to perform immunostaining in control presumed normal embryos. This has consequences on some of the conclusions in our study. For example, phospho-S15-p53 was high in the arrested embryos (**Fig 3G**) and potentially it might be high in normal embryos; however, high phospho-S15-p53 has not been reported in human or other species embryos. Secondly, most of the data presented here takes the average of single embryos, which will blur gene expression signatures. However, the MZT has been successfully characterized from single-embryo RNA-seq data [4], and RNA-seq is capable of detecting rare transcripts and subtle changes in cell type in bulk samples [20], for example, the level of differentiation in human ESCs can be estimated from bulk RNA-seq [80].

Quiescence and senescence describe the ability of some cells to arrest and exit the cell cycle. This phenomenon is best explored in somatic cells. Stress signals, such as UV irradiation, or cellular oxidative stress led to the deregulation of cell cycle regulatory pathways. Sometimes these activations result in senescence, which is irreversible. While active cycling and quiescence are normal cellular responses, senescence is associated with ageing and with disease [50,81]. There are, however, differences between the arrested state in the embryos and senescence in somatic cells. We observed increased p21 (*CDKN1A*) and phospho-S15-p53, but *FOXO1* expression was not up-regulated, although FOX-family TFs appear to be more active in the arrested embryos than the normal embryos. The down-regulation of MYC activity is reminiscent of the induction of diapause in mouse blastocysts [78]. Indeed, there are parallels between the MYC-induced diapause-like state and the arrested embryos: In both cases, ribosome expression is reduced, cell cycle activity declines. However, diapause occurs at the blastocyst stage, while the arrest seen here is pre-compaction, suggesting the biological process is different.

Aneuploidies have been suggested to contribute to embryonic arrest. Human embryos have high levels of aneuploidy, and as much as 73% of blastocysts are mosaic [27,82,83]. However, while aneuploidy is highly deleterious for later postimplantation development [84], there is mixed evidence before the blastocyst stage [34]. Indeed, aneuploid human embryos still reached the blastocyst stage [85], and whole chromosome aneuploidies are mainly lethal post-implantation [34]. Although, a screen for aneuploidy in human morula/blastocysts suggests human embryos with very high levels of mosaicism are rare [33], suggesting a mechanism that culls severe mosaic aneuploidies before compaction [30]. Nonetheless, while we cannot rule it out, our data suggest that aneuploidy is not a dominant mechanism behind the embryonic arrest of cleavage-stage embryos.

We show that some arrested embryos can be induced to recommence development through the application of 2 small molecules, resveratrol and NR, both can activate the sirtuin class of deacetyltransferases. We speculate that these 2 drugs are altering the metabolic balance of the embryos and are acting to push the embryos out of their senescent-like state. Metabolism in human embryos is initially hypoxic and relies upon oxidative phosphorylation using pyruvate. Indeed, human IVF grown embryos are sensitive to high levels of oxygen [86], and blastulation rates are increased when oxygen was kept at 5% during embryo culture [87] (similar conditions were used in this study). After implantation, glycolysis becomes the dominant metabolic pathway. Resveratrol activates SIRT1, which has been reported to inhibit glycolysis in somatic cells. However, paradoxically resveratrol-treated embryos had enhanced glycolytic pathways.

There is some evidence to support a different role of SIRT1 in embryonic cells. For example, Cha and colleagues showed that high SIRT1 expression is a feature of hPSCs that are primarily glycolytic [67,88]. Cha and colleagues speculated that SIRT1 may be exerting these effects not by directly regulating glycolytic enzymes, as occurs in somatic cells, but through an indirect process involving SIRT2 acetylation of glycolytic enzymes [67,89]. We speculate that SIRT1 is regulating fatty acid oxidation but not glycolysis in embryos, but the balance of these 2 processes is crucial for the transition to the glycolytic environment in early implantation. Resveratrol treatment led to changes in fatty acid oxidation, which agrees with previous studies showing a role for SIRT1 [71,72]. Our data suggest that the arrested embryos are failing to up-regulate glycolytic-based or fatty acid oxidation metabolism.

Resveratrol and NR can reactivate development in a limited number of cases; however, the reactivated embryos continue to show problems, such as persistent down-regulation of ribosomes and translation genes. A possible explanation is that resveratrol is forcing the embryos into an unnatural state that reactivates development, but cellular problems persist. Overall, our data indicate 2 primary mechanisms to explain arrested embryos (**Fig 6D**): A failure to correctly traverse the MZT (40% of arrested embryos) and a failure to regulate metabolic pathways (60% of arrested embryos).

## Materials and methods

### Human embryo collection

All patients received standard antagonist protocol for ovarian stimulation [90]. Patients were injected with recombinant FSH (Gonal-F, Merck, Italy) on day 2 of their menstrual cycle with a starting dose of 150 IU/d. Transvaginal ultrasound and blood E2 levels were used to monitor follicle growth. When at least 2 follicles grew larger than 18 mm in diameter, 0.1 mg of gonadotropin releasing hormone agonist (Triptorelin, Ferring GmbH, Germany) and 4,000 IU of human chorionic gonadotropin was injected as a trigger. Approximately 36 h after trigger, transvaginal ultrasound-guided oocyte retrieval was performed under anesthesia.

### In vitro fertilization of oocytes and culture of embryos

Fertilization and in vitro culture procedures were performed as previously described [91]. Briefly, oocyte cumulus complexes were identified and washed in G-IVF (Vitrolife, Sweden), then inseminated with 50,000 to 100,000 normal motile spermatozoa, oocytes were examined for successful fertilization after 16 to 18 h, zygotes with 2 pronuclei were allowed to continue culture for 48 h in G1 PLUS medium. On day 3, embryos were observed for morphology and blastomere numbers, embryos with 2 to 5 cells at day 3 were considered as possible arrested embryos. Arrested embryos were cultured in G2 PLUS medium for a further day (Vitrolife, Sweden), and if there was no further cell division then the embryos were defined as arrested. Arrested embryos were left untreated or treated with solvent, resveratrol (MCE, United States of America) or nicotinamide riboside/NR (Sigma-Aldrich, USA), in the resveratrol group, embryos were cultured with 1 μm resveratrol for 6 h, and then transferred into fresh G2 PLUS medium for 42 h. In the NR-treated embryos, embryos were cultured with 1 mM NR for 24 h, and then transferred into fresh G2 PLUS for 24 h; in the control group, embryos were cultured with fresh G2 PLUS for 48 h. Embryos were also treated with 0.5 μm rapamycin (Sigma-Aldrich), 0.5 μm PD0325901 (Selleck), and 25 μg/ml vitamin C (Sigma-Aldrich), for 24 h, then transferred into fresh G2 PLUS medium for a further 24 h. The morphology and blastomere numbers were observed on days 4 and 5. Embryos were cultured in a tri-gas incubator with an environment of 6% $CO_2$, 5% $O_2$, and 89% $N_2$ at 37˚C.

## Immunofluorescence

After culture, embryos were fixed with 4% (w/v) paraformaldehyde for 30 min at room temperature, followed by washing in PBS (phosphate-buffered saline) for 3 times. Embryos were permeabilized in 0.5% Triton X-100 for 30 min at room temperature, and blocked with blocking buffer (Beyotime, China) for 1 h at 37°C. Then, embryos were then incubated with anti-SIRT1 antibody (1:100, Abcam, #ab189494, USA), anti-NRF2 antibody (1:100, Abcam, #ab137550, USA), anti-phospho-AKT (Ser473) antibody (1:200, CST, #4060S, USA), anti-phospho-P53 (Ser15) antibody (1:400, CST, #82530s, USA), or anti-p21 antibody (1:100, Abcam, #ab109520 USA) at 4°C overnight. Embryos were subsequently incubated with Alexa-Fluor-488 secondary antibody (1:500, Abcam, # ab150077, USA) for 2 h at 37°C, then washed 3 times. The DNA was stained with 4,6-diamino-2-phenyl indole (DAPI). Finally, the embryos were suspended in a microdrop with blocking solution and photographed with a fluorescence microscope (MetaSystems, Germany).

## Single-embryo RNA-seq

Single-embryo RNA-seq was performed essentially as described [92]. Briefly, single embryos were placed into 10 μl of SMART-seq2 lysis buffer (1 μl RNAse inhibitor, 0.2% (v/v) Triton X-100) and stored at −80°C before sequencing. Embryos were sequenced on an Illumina sequencer according to the manufacturer's instructions.

## RNA-seq data analysis

RNA-seq data were analyzed essentially as described in [20], with the exception that the STAR aligner was used [93], RSEM was replaced with scTE [21], and GENCODE v32 [94] was used for the transcript annotations. Data were GC-content normalized with EDASeq [22]. Differential gene expression was determined using DESeq2 [95], genes were defined as significantly DE if they changed by at least 4-fold, and had a q-value (Benjamini–Hochberg corrected *p*-value) of 0.01 or less. GSEA was performed using fgsea, and all of the GSEAs shown in the manuscript had a q-value (Bonferroni–Hochberg corrected *p*-value, from 10,000 permutations) of less than 0.05 [96]. CytoTRACE [25] was performed using default parameters on the unnormalized raw tag count matrix (https://figshare.com/articles/dataset/Human_embryo_normalized_gene_expression_data/19775992). Karyotype was estimated from the RNA-seq data using the software from https://github.com/MarioniLab/Aneuploidy2017 [31]. The presumed normal samples are from a reanalysis of GSE66507 [18], PRJEB11202 [19], and GSE36552 [17] embryo RNA-seq data. Other analysis was performed using glbase3 [97].

## Statistics

Statistical analysis was used for the differential expression measures. Differential expression was determined using DESeq2 [95], and minimum thresholds of >4-fold change, and a q-value of 0.01 (Benjamini–Hochberg corrected *p*-value) was used to determine significantly different. For GSEA, a minimum q-value of 0.05 (Benjamini–Hochberg corrected *p*-value) was considered significant, as estimated by random permutation by fgsea [96]. Significance was also calculated from a 2-sided Welch's *t* test for sets of genes.

## Study approval

This study was approved by Ethical Approval Board (Approval number: 2020-866-75-01) of the Shuguang Hospital affiliated to Shanghai University of Traditional Chinese Medicine and Southern University of Science and Technology ethical committee (Approval number:

2021SWX012). Arrested embryos used in this study were rejected embryos from normal IVF procedures and were used with the patients informed, written consent. No normal control embryos (non-arrested) were used in this study.

## Supporting information

**S1 Data. Values for several bar charts and figures in the manuscript, including: Raw data for Figs 1A, 1B, 2H, 3B, 3C, 4A, 4B, 5D, 5E, 5H, 5I, 5J, S5A, S7A, S7C and S9A.**
(XLSX)

**S2 Data. Karyotype calculation result.**
(XLSX)

**S3 Data. All differentially regulated MZT genes, including gene rankings for GSEA plots.**
(XLSX)

**S4 Data. All differentially regulated genes/TEs identified in this study.**
(XLSX)

**S5 Data. Gene ranks for GSEA plots for all comparisons in this study.**
(XLSX)

**S1 Fig. (Related to Fig 2). Arrested embryos maintain developmental potential. (A)** Cyto-TRACE cell embedding manifold, colored by cell type and embryo stage (left) or by predicted developmental order (right). A predicted developmental order score of 1.0 is less differentiated, and a score of 0.0 is more. E = Embryonic-stage samples, as defined in [19], for this and all subsequent figures. (**B**) Box plots of the predicted ordering from CytoTRACE for all cells/embryos at the indicated stages, ordered by the mean developmental predicted ordering. Each dot is a cell/embryo. (**C**) As in **panel B**, but only showing the stages from oocyte to morula. (**D**) Violin plots showing expression of 4-cell/8-cell-specific human genes. Underlying data can be found in: https://figshare.com/articles/dataset/Human_embryo_normalized_gene_expression_data/19775992. (**E**) Violin plots showing expression of 8-cell/E4-specific human genes, i.e., developmental genes involved in the establishment of the blastocyst. Underlying data can be found in: https://figshare.com/articles/dataset/Human_embryo_normalized_gene_expression_data/19775992. (**F**) Heatmap of the Z-scores of expressions of the genes in the 8-cell signature identified in [26]. Underlying data can be found in: https://figshare.com/articles/dataset/Human_embryo_normalized_gene_expression_data/19775992. All of the panels derived from CytoTRACE use non-normalized tag counts from: https://figshare.com/articles/dataset/Human_embryo_normalized_gene_expression_data/19775992.
(PDF)

**S2 Fig. (Related to Fig 2). Arrested embryos are (mainly) karyotypically normal. (A)** Karyotype abnormalities estimated using the approach outlined in [31]. In these plots, each chromosome is plotted separately, and the gray dots are each cell/embryo in the indicated stage of development or arrest. Red dots indicate when the gene expression on that chromosome exceeds the Z-score threshold and is significantly over or under represented. Red dots are indicative of aneuploidies, and those above the line suggest a gain of a chromosome or part of a chromosome, and those below the line suggest a loss of a chromosome or part. (**B**) Bar chart showing the predicted aneuploidies in the indicated developmental stages. Red bars indicate cells/embryos with a predicted loss or gain of a chromosome, while those in gray are predicted to be normal. (**C**) Percentage of the aneuploidies observed, broken down by chromosome in the normal embryo dataset. (**D**) As in **panel C**, but only the arrested embryos. All of these

panels use data underlying results from **S2 Data**.
(PDF)

**S3 Fig. (Related to Fig 2). Type I arrested embryos have MZT problems.** (**A**) Volcano plot showing the fold-change versus significance when comparing 2-cell-stage embryos versus 8-cell-stage. Differential expression was calculated using DESeq2, and a minimum fold-change of 4, and a q-value of 0.01 was considered significantly different. The q-value is the Bonferroni–Hochberg multiple test corrected *p*-value. We defined "maternal RNA-clearance genes" as those that were significantly down-regulated, and "major ZGA genes" as those that were up-regulated. Significantly DE up-regulated genes are labeled in red and down-regulated genes in blue. The number of genes passing the differential expression thresholds are marked on the plot. DE genes/TEs are listed in **S3 Data**. (**B**) GSEA for the up- and down-regulated genes as ranked in **panel A**. Underlying data can be found in **S3 Data**. (**C**) Violin plots for the expression of key major ZGA genes, *DUX4*, *DUXA*, and *ZSCAN4*. Significance is from a 2-sided Welch's *t* test. Underlying data can be found in: https://figshare.com/articles/dataset/Human_embryo_normalized_gene_expression_data/19775992. (**D**) Violin plots for the expression of critical maternal RNA clearance genes *CNOT6L* and *PAN2*. Significance is from a 2-sided Welch's *t* test. Underlying data can be found in: https://figshare.com/articles/dataset/Human_embryo_normalized_gene_expression_data/19775992. DE, differentially expressed; GSEA, gene set enrichment analysis; MZT, maternal-to-zygotic transition; TE, transposable element; ZGA, zygotic genome activation.
(PDF)

**S4 Fig. (Related to Fig 2). Transposable element expression is disturbed in Type I arrested embryos.** (**A**) Volcano plot for all genes when comparing Type I arrested embryos to 2-cell-stage cells. Differential expression was calculated using DESeq2, and a minimum fold-change of 4, and a q-value of 0.01 was considered significantly different. The q-value is the Bonferroni–Hochberg multiple test corrected *p*-value. Significantly DE up-regulated genes are labeled in red and down-regulated genes in blue. The number of genes passing the differential expression thresholds are marked on the plot. DE genes are listed in **S4 Data**. (**B**) Volcano plot, as in **panel A**, but only containing TE types, and comparing Type I arrested embryos to 2-cell-stage embryos (left volcano), or 8-cell-stage embryos (right volcano). DE TEs are listed in **S4 Data**. (**C**) Volcano plot, as in **panel A**, but only containing TE types, and comparing Type II arrested embryos to morula-stage embryos (left volcano), or Type III arrested embryos versus E4 (early blastocyst)-stage embryos (right volcano). DE TEs are listed in **S4 Data**. (**D**) Number of differentially regulated TE types for the indicated TEs, when comparing Type I arrested embryos to 4-cell-stage embryos. DE TEs are listed in **S4 Data**. (**E**) Heatmaps of the RNA-seq read tag density for all genomic copies of LINE L1HS copies (rows). Heatmaps are the density of normalized tag counts (in reads per million) for each sample and are ranked by the sum of each row for each heatmap. The raw data for this plot is available from GSA under the accession HRA001406. The genome locations for the LINE L1HS are available for download from the UCSC genome browser. (**F**) Heatmap for the expression of all differentially regulated TEs in the Type I arrested embryos. The rows are the 315 TEs identified in panel B, the columns are the arrested embryos and all stages form Oocyte to late blastocyst. The location of the Type I embryos is indicated on the top row. DE TEs are listed in **S4 Data**. DE, differentially expressed; TE, transposable element.
(PDF)

**S5 Fig. (Related to Fig 3). Arrested embryos have reduced ribosome and nucleosome expression.** (**A**) Number of significantly DE genes (top chart) and TEs (bottom chart) (a fold-

change of at least 4, and a Bonferroni–Hochberg corrected q-value of less than 0.01) in the indicated comparisons. Underlying data can be found in **S1 Data**. (**B**) Volcano plots showing all genes when comparing Type II versus morula (left) or Type III versus E4 (right)-stage embryos. Differential expression was calculated using DESeq2, and a minimum fold-change of 4, and a q-value of 0.01 was considered significantly different. The q-value is the Bonferroni–Hochberg multiple test corrected *p*-value. Significantly DE up-regulated genes are labeled in red and down-regulated genes in blue. The number of genes passing the differential expression thresholds are marked on the plot. DE genes are listed in **S4 Data**. (**C**) Violin plot for the expression of selected histones. Significance is from a 2-sided Welch's *t* test. Underlying data can be found in: https://figshare.com/articles/dataset/Human_embryo_normalized_gene_expression_data/19775992. n.s. = not significant. (**D**) Violin plot for selected large or small ribosome subunits. Significance is from a 2-sided Welch's *t* test. Underlying data can be found in: https://figshare.com/articles/dataset/Human_embryo_normalized_gene_expression_data/19775992. (**E**) Heatmap of the expression of all histones/nucleosomes and selected senescence-related genes (from the REACTOME category: SENESCENCE–ASSOCIATED SECRETORY PHENOTYPE (SASP): REACTOME: R–HSA–2559582.2). Expression is presented as log2 NTC. Underlying data can be found in: https://figshare.com/articles/dataset/Human_embryo_normalized_gene_expression_data/19775992. Heatmap of all significantly DE (fold-change >4 and q-value <0.01) large ribosome subunits. Expression is presented as log2 NTC. Underlying data can be found in: https://figshare.com/articles/dataset/Human_embryo_normalized_gene_expression_data/19775992. (**F**) As in **panel E**, but for all significantly DE small ribosome subunits. Underlying data can be found in: https://figshare.com/articles/dataset/Human_embryo_normalized_gene_expression_data/19775992. DE, differentially expressed; NTC, normalized tag count; TE, transposable element.
(PDF)

**S6 Fig. (Related to Fig 3). Arrested embryos have decreased expression of cell cycle genes.** (**A**) Heatmap showing the expression of selected cell cycle–related genes. The cell/embryo stage is indicated in the top, colored bar legend. Genes specifically expressed in G1/S or G2/M to a cell cycle phase (as defined in [48]) are marked on the right-hand side of the heatmap. Underlying data can be found in: https://figshare.com/articles/dataset/Human_embryo_normalized_gene_expression_data/19775992. (**B**) Violin plots showing expression of cell cycle–related gene, *CCNB1*, *PCNA*, and the tubulin subunits *TUBA1A* and *TUBB4B*. Significance is from a 2-sided Welch's *t* test. Underlying data can be found in: https://figshare.com/articles/dataset/Human_embryo_normalized_gene_expression_data/19775992. (**C**) Violin plot for the expression of *TP53* (p53). Significance is from a 2-sided Welch's *t* test. Underlying data can be found in: https://figshare.com/articles/dataset/Human_embryo_normalized_gene_expression_data/19775992. n.s. = not significant. (**D**) Heatmap for the expression of p53 target genes (HALLMARK_P53_PATHWAY set), ranked by the sum of the columns. Each column is a single cell or embryo, and the arrested embryos are labeled in pink. Underlying data can be found in: https://figshare.com/articles/dataset/Human_embryo_normalized_gene_expression_data/19775992.
(PDF)

**S7 Fig. (Related to Fig 4). Treatment of arrested embryos with small molecules, and resveratrol corrects developmental and cell cycle problems, but not ribosomes and nucleosome expression.** (**A**) Percentage of arrested embryos that recommenced development, and the stage they reached, when treated with the indicated small molecules. Underlying data can be found in **S1 Data**. (**B**) Violin plot showing the expression of the blastocyst-related genes *ESRRB* and *TFCP2L1* in the indicated embryonic stages and in arrested and resveratrol-treated

embryos. Underlying data can be found in: https://figshare.com/articles/dataset/Human_ embryo_normalized_gene_expression_data/19775992. (**C**) Violin plots showing the distribution of Z-scores of expression for all the small and large ribosomes and histone genes in the indicated embryonic stages and in arrested and resveratrol-treated embryos. Significance is from a 2-sided Welch's *t* test. Underlying data can be found in: https://figshare.com/articles/ dataset/Human_embryo_normalized_gene_expression_data/19775992. (**D**) Violin plots showing the expression of the cell cycle–related genes *PCNA* and *CCNA2* in the indicated embryonic stages and in arrested and resveratrol-treated embryos. Significance is from a 2-sided Welch's *t* test. Underlying data can be found in: https://figshare.com/articles/dataset/ Human_embryo_normalized_gene_expression_data/19775992. (**E**) Volcano plot showing all significantly differentially expressed (fold-change >4 and q-value <0.01) for all genes and TEs when comparing resveratrol-treated versus Type I arrested embryos. Differential expression was calculated using DESeq2, and a minimum fold-change of 4, and a q-value of 0.01 was considered significantly different. The q-value is the Bonferroni–Hochberg multiple test corrected *p*-value. Significantly DE up-regulated genes are labeled in red and down-regulated genes in blue. The number of genes passing the differential expression thresholds are marked on the plot. Underlying data can be found in **S4 Data**. (**F**) Plot showing the expression ranks of TEs ordered by their expression. TEs for each embryo or cell were ranked by their total expression and the curves were plotted. Normal embryos are indicated in gray ("Other"), and Types I–III and resveratrol are in the indicated colors. Underlying data can be found in: https://figshare. com/articles/dataset/Human_embryo_normalized_gene_expression_data/19775992. DE, differentially expressed; TE, transposable element.
(PDF)

**S8 Fig. (Related to Fig 5). Comparison of resveratrol-treated versus other embryonic cells.** (**A**) Volcano plot showing all significantly differentially expressed (fold-change >4 and q-value <0.01) for all genes and TEs when comparing showing resveratrol versus Type II (left) or Type III (right) arrested embryos. Differential expression was calculated using DESeq2, and a minimum fold-change of 4, and a q-value of 0.01 was considered significantly different. The q-value is the Bonferroni–Hochberg multiple test corrected *p*-value. Significantly DE up-regulated genes are labeled in red and down-regulated genes in blue. The number of genes passing the differential expression thresholds are marked on the plot. DE genes/TEs are listed in **S4 Data**. (**B**) GSEA showing significantly different terms for resveratrol versus Type II arrested embryos. Underlying data can be found in **S5 Data**. (**C**) Violin plots showing the expression of the indicated glycolysis-related genes *ALDH1B1*, *GAPDH*, *PGM1*, and *PFKL*. Significance is from a 2-sided Welch's *t* test. Underlying data can be found in: https://figshare.com/articles/ dataset/Human_embryo_normalized_gene_expression_data/19775992. n.s. = not significant. (**D**) Volcano plots (as in **panel A**), but showing resveratrol-treated embryos versus E5-stage embryos. DE genes are listed in **S4 Data**. (**E**) GSEA of the ranked DE genes from **panel D**. Underlying data can be found in **S5 Data**. DE, differentially expressed; GSEA, gene set enrichment analysis; TE, transposable element.
(PDF)

**S9 Fig. (Related to Fig 5). Resveratrol up-regulates the expression of fatty acid metabolic genes.** (**A**) 2D dot plot showing the sum of the Z-scores for the genes in the indicated KEGG categories. The x-axis scores the fatty acid metabolism pathway, the y-axis the oxidative phosphorylation pathway. Each dot in the plot is a cell/embryo, and the top and right axis have the kernel density for each group of cells. The resveratrol-treated embryos have a filled in color (pink) for emphasis. The arrested and resveratrol-treated embryos are indicated by dashed lines, and the normal developmental states are indicated by labels. Underlying data can be

found in **S1 Data**. (**B**) Violin plots showing the expression of selected fatty acid-related metabolic genes. Significance is from a 2-sided Welch's *t* test. Underlying data can be found in: https://figshare.com/articles/dataset/Human_embryo_normalized_gene_expression_data/19775992. (**C**) A model for the action of resveratrol and NR on SIRTs and metabolic pathways. The dotted arrow between SIRT1 and glycolysis implies indirect regulation. NR, nicotinamide riboside.
(PDF)

**S10 Fig. (Related to Fig 6). Transcriptional regulation of embryonic arrest.** (**A**) All significantly enriched motifs in the Types I–III arrested embryos in the promoters of DE genes. Motif discovery was performed using HOMER with default settings [98], against the promoters (defined here as –1,000 bp upstream) of the DE genes in the indicated types of arrested-embryo. The size of the circle indicates the percent of gene promoters that had the motif, and the color indicates the *p*-value for enrichment. (**B**) Violin plot showing the expression of *FOXO1* in the indicated embryonic cell types. Significance is from a 2-sided Welch's *t* test. Underlying data can be found in: https://figshare.com/articles/dataset/Human_embryo_normalized_gene_expression_data/19775992. n.s. = not significant. DE, differentially expressed.
(PDF)

## Author Contributions

**Conceptualization:** Guoqing Tong.

**Data curation:** Yang Yang, Xiuling Fu, Ye Xia, Xiufang Zhong.

**Formal analysis:** Liyang Shi, Xiuling Fu, Gang Ma, Zhongzhou Yang, Yuhao Li, Yibin Zhou, Lihua Yuan, Ye Xia, Xiufang Zhong, Ping Yin, Li Sun, Wuwen Zhang, Isaac A. Babarinde, Yongjun Wang, Andrew P. Hutchins.

**Investigation:** Yang Yang, Gang Ma.

**Project administration:** Andrew P. Hutchins, Guoqing Tong.

**Resources:** Andrew P. Hutchins, Guoqing Tong.

**Software:** Liyang Shi.

**Supervision:** Wuwen Zhang, Andrew P. Hutchins, Guoqing Tong.

**Validation:** Yuhao Li, Ye Xia, Xiufang Zhong, Li Sun, Wuwen Zhang, Yongjun Wang, Xiaoyang Zhao.

**Writing – original draft:** Andrew P. Hutchins.

**Writing – review & editing:** Yibin Zhou, Isaac A. Babarinde, Xiaoyang Zhao, Andrew P. Hutchins, Guoqing Tong.

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
