## [Editor Report · Decision Letter 0]

23 Jan 2022

Dear Dr Hutchins, 

Thank you for submitting your manuscript entitled "Human embryos arrest in a quiescent-like state characterized by metabolic and zygotic genome activation problems" for consideration as a Research Article by PLOS Biology. Thank you also for your patience as we completed our editorial process, and please accept my apologies for the delay in providing you with our decision.

Your manuscript has now been evaluated by the PLOS Biology editorial staff as well as by an academic editor with relevant expertise and I am writing to let you know that we would like to send your submission out for external peer review.

Once your full submission is complete, your paper will undergo a series of checks in preparation for peer review. Once your manuscript has passed the checks it will be sent out for review. To provide the metadata for your submission, please Login to Editorial Manager (https://www.editorialmanager.com/pbiology) within two working days, i.e. by Jan 25 2022 11:59PM.

If your manuscript has been previously reviewed at another journal, PLOS Biology is willing to work with those reviews in order to avoid re-starting the process. Submission of the previous reviews is entirely optional and our ability to use them effectively will depend on the willingness of the previous journal to confirm the content of the reports and share the reviewer identities. Please note that we reserve the right to invite additional reviewers if we consider that additional/independent reviewers are needed, although we aim to avoid this as far as possible. In our experience, working with previous reviews does save time. 

If you would like to send previous reviewer reports to us, please email me at ialvarez-garcia@plos.org to let me know, including the name of the previous journal and the manuscript ID the study was given, as well as attaching a point-by-point response to reviewers that details how you have or plan to address the reviewers' concerns. 

Given the disruptions resulting from the ongoing COVID-19 pandemic, please expect some delays in the editorial process. We apologise in advance for any inconvenience caused and will do our best to minimize impact as far as possible.

Kind regards,

Ines

--

Ines Alvarez-Garcia, PhD

Senior Editor

PLOS Biology

---

## [Decision Letter · Decision Letter 1]

5 Mar 2022

Dear Dr Hutchins,

Thank you for submitting your manuscript entitled "Human embryos arrest in a quiescent-like state characterized by metabolic and zygotic genome activation problems" for consideration as a Research Article at PLOS Biology. Thank you also for your patience as we completed our editorial process, and please accept my apologies for the delay in providing you with our decision. Your manuscript has been evaluated by the PLOS Biology editors, an Academic Editor with relevant expertise, and by two independent reviewers.

As you will see, the reviewers find your conclusions interesting and the manuscript worth pursuing for publication. Nevertheless, they also raise several points that need to be addressed to strengthen the results. Reviewer 1 suggests several approaches to determine the cell cycle state of the cells within an arrested embryo and Reviewer 2 thinks you should discuss further the possible reason why the human embryos are arrested early during embryogenesis and asks for the clarification of several issues.

In light of the reviews (attached below), we are pleased to offer you the opportunity to address the comments from the reviewers in a revised version that we anticipate should not take you very long. We will then assess your revised manuscript and your response to the reviewers' comments and we may consult the reviewers again.

We expect to receive your revised manuscript within 1 month.

**IMPORTANT - SUBMITTING YOUR REVISION**

3. Resubmission Checklist

a) *PLOS Data Policy*

Please note that as a condition of publication PLOS' data policy (http://journals.plos.org/plosbiology/s/data-availability) requires that you make available all data used to draw the conclusions arrived at in your manuscript. If you have not already done so, you must include any data used in your manuscript either in appropriate repositories, within the body of the manuscript, or as supporting information (N.B. this includes any numerical values that were used to generate graphs, histograms etc.).For an example see here: http://www.plosbiology.org/article/info%3Adoi%2F10.1371%2Fjournal.pbio.1001908#s5

Please provide the data underlying the graphs of the following figures:

Fig. 1A, B; Fig. 2A-I: Fig. 3A-J; Fig. 4A, B, D, E; Fig. 5A, D-I; Fig. 6B, C; Fig. S1A-E; Fig. S2A-D; Fig. S3A-D; Fig. S4A-E; Fig. S5A-G; Fig. S6A-E; Fig. S7A-D; Fig. S8A-E; Fig. S9A, B and Fig. S10B

Please also indicate in each figure legend WHERE THE DATA CAN BE FOUND. 

b) *Published Peer Review*

Sincerely,

Ines

--

Ines Alvarez-Garcia, PhD

Senior Editor

PLOS Biology

Reviewers' comments

Rev. 1:

In the manuscript of Yang et al., the authors performed RNA-seq of arrested IVF human embryos in order to identify potential causes of developmental arrest. By comparing their RNA-seq data with published scRNA-seq of early human embryos, the authors classified the arrested embryos to three types according to their expression signature. They further demonstrated that the treatment of resveratrol or nicotinamide riboside partially rescued the embryo arrest through metabolic pathways. Overall, this is an interesting study providing potential explanations for IVF embryo arrest, however, detailed mechanistic study and additional controls are needed to support the conclusions.

1. In line 18, single-embryo RNA-seq was used to identify the expression profiles, but multiple cells are present in a single arrested embryo (Figure 1). The developmental stage of each cell in an arrested embryo and the causes of the developmental arrest may be all different among cells. So the single-embryo RNA-seq only provide average expression signals from all cells within one embryo. So it is not accurate to classify the types of embryo arrests based on an averaged signal of different cells.

2. Cell number is correlated with developmental stages during pre-implantation development. Are there any relationship between the number of cells in arrested embryo and the arrested stage? Type I arrested embryos expressed 4/8-cell marker genes (line 145-146), but the authors claimed that Type I arrested embryos were at 2/4-cell stage according to Figure 2A. Please explain the reason that Type I arrested embryos are not assigned to 4/8-cell stage.

3. In Figure 2C, the authors claimed that ZGA genes were incompletely activated and maternal transcripts were not cleared. Given that each arrested embryo may contain cells of different developmental stages, the observed gene expression defects could be due to an averaged signal of a mixture of cells from different developmental stages instead of inefficient MZT in type I arrested embryos.

4. In Figure 2H-I, the authors showed that TEs were activated but ZGA failed in type I arrested embryos. Are these activated TEs expressed in other developmental stages? What are the functions of abnormally activated TEs in type I arrested embryos and during development? Are the expression of TEs reversed by the addition of resveratrol or NR?

5. Figure 2H showed the activation of TEs in type III embryos, but Figure S4C and E showed otherwise. Could the authors explain? Is the activation of TEs in type I/III arrested embryos statistically significant in Figure 2H? The change of TEs may be unrelated to gene expression or ZGA. So it is necessary for the authors show expression for genes only (not genes and TEs together) between type I/II/III arrested embryos and embryos of different developmental stages.

6. Figure 3D only showed GSEA for senescence genes, not quiescence genes. The authors should not claim quiescence (line 293-294).

7. The authors showed that lower percentage of embryos located within G2/phase (Figure 3E). Again, even within the same embryo, each cell may be at different cell cycle stage. The authors cannot draw conclusions for cell cycle of arrested embryos based on current RNA-seq data. Other experimental approaches should be used to determine the cell cycle state of cells within an arrested embryo.

8. The authors claimed that resveratrol rescued the arrested embryos, but why "the resveratrol-treated embryos still have deregulated levels of ribosomes, protein translation" and quiescence markers? This weakens the claim that cell cycle quiescence is the reason for developmental arrest.

9. "Resveratrol was pushing cells towards an increased glycolytic metabolism, whereas all the arrested cells had reduced glycolysis" (line 426-428) and Resveratrol activates Sirt1. But Sirt1 was reported to inhibit glycolysis, how to explain the discrepancy?

10. The authors should upload all their single embryo RNA-seq data to public database and provide the link for reviewers to access. It is hard to judge the quality of the sequencing data and conclusions without accessible data.

11. Important controls are missing. Morphology in control embryos should be added in Figure 1C and 4C. Immunostaining in control embryos should be added in 3I, 4F-G and 5B-C. Treating arrested embryos with solvent of Resveratrol and NR should be added as control.

12. Statistical analysis is lack, in all the figures, e.g. Figure 2D-F, 3A-F, 3G-J, 4E, 5A and all GSEA figures. NES values and FDR values are lack in all GSEA figures.

Rev. 2:

In this manuscript, the authors investigated the molecular mechanism responsible for developmental arrest at different developmental stages during pre-implantation development of human embryos. They collected human embryos arrested at different stages and then performed single-cell RNA sequencing and bioinformatic analysis to further dissect the molecular events. By analyzing the RNA-seq data, they divided the arrested embryos into three types. The Type I embryos were arrested possible because of failure in zygotic genome activation. While the Type II and III embryos were arrested because of cell cycle progression and metabolic problem. Finally, they found that the supplementation of resveratrol or nicotinamide riboside (NR) in the culture medium can partially rescue the arrested phenotype observed. Overall, the finding is interesting and should be publishable after revision.

1. They observed that ZGA is defective in Type I arrested embryos, which is predictable since these embryos were arrested at 2-5 cell stage and major ZGA occurred at 8-cell stage in human. Since the authors already found that ZGA genes were not activated and maternal genes expression remained very high, they can further discuss the possible reason causing this early stage arrest in human embryos.

2. Both Type II and III arrested embryos exhibited cell cycle and metabolic defects, and it will be difficult for the reviewer to understand why the authors divided these arrested embryos into two types because they further presented their results together and no obvious difference can be observed anyway.

---

## [Decision Letter · Decision Letter 2]

13 May 2022

Dear Dr Hutchins,

Thank you for your patience while we considered your revised manuscript entitled "Human embryos arrest in a senescent-like state characterized by metabolic and zygotic genome activation problems" for publication as a Research Article at PLOS Biology. This revised version of your manuscript has been evaluated by the PLOS Biology editors, the Academic Editor and one of the original reviewers.

Based on the review (attached below) and discussions with the team of editors, we are likely to accept this manuscript for publication, provided you address a few minor editorial requests:

1) We think the title can be improved and we would like you to consider one of the suggestions below:

"Metabolic and epigenetic dysfunctions underlie the arrest of in vitro fertilized human embryos in a senescent-like state"

"The frequently-observed arrest of in vitro fertilized human embryos is associated with metabolic and epigenetic dysfunction that triggers a senescent-like state"

"The frequently-observed arrest of vitro fertilized human embryos is associated with a senescent-like state that is characterized by metabolic and epigenetic dysfunction."

2) Please note that in Fig. 3 legend, you skipped the letter ‘H’ and section ‘J’ should be named ‘H’

3) Regarding the data deposited at Genome Sequence Archive (HRA001406) please remember that the data should be made available at publication time.

We expect to receive your revised manuscript within two weeks. 

*Published Peer Review History*

*Press*

Sincerely,

Ines

--

Ines Alvarez-Garcia, PhD

Senior Editor

PLOS Biology

Reviewers' comments

Rev. 1: Xinyi Lu - note that this reviewer has signed his review

The authors have addressed all my comments and questions from the previous review and made significant improvements to the manuscript. I will support the publication of the manuscript.

---

## [Editor Report · Decision Letter 3]

19 May 2022

Dear Dr Hutchins,

On behalf of my colleagues and the Academic Editor, Christa Bücker, I am happy to say that we can in principle accept your Research Article entitled "Metabolic and epigenetic dysfunctions underlie the arrest of in vitro fertilized human embryos in a senescent-like state" for publication in PLOS Biology, provided you address any remaining formatting and reporting issues. These will be detailed in an email that will follow this letter and that you will usually receive within 2-3 business days, during which time no action is required from you. Please note that we will not be able to formally accept your manuscript and schedule it for publication until you have completed any requested changes.

PRESS

Many congratulations for your paper and thanks again for choosing PLOS Biology for publication and supporting Open Access publishing. We look forward to publishing your study. 

Sincerely, 

Ines

--

Ines Alvarez-Garcia, PhD 

Senior Editor 

PLOS Biology
